# Enhancing Sulfate Erosion Resistance in Ultra-High-Performance Concrete through Mix Design Optimization Using the Modified Andreasen and Andersen Method

Guan Wang [1], Wenlin Chen [2,*], Xiangyu Shen [2], Xin Ren [2], Jiawei Niu [3], Sihang Pan [2], Yifan Huang [4] and Jinliang Wu [2]

1   Chongqing Jiaoda Construction Engineering Quality Test Center Co., Ltd., Chongqing 400074, China; 15922705053@163.com
2   School of Civil Engineering, Chongqing Jiaotong University, Chongqing 400074, China; 18660603851@163.com (X.S.); 622220951009@mails.cqjtu.edu.cn (X.R.); p17623363058@163.com (S.P.); jinliangwu@126.com (J.W.)
3   Chongqing Academy of Science and Technology, Chongqing 401121, China; niujw2018@163.com
4   School of Transportation, Southeast University, Nanjing 211189, China; huangyifan@seu.edu.cn
*   Correspondence: 622210951032@mails.cqjtu.edu.cn; Tel.: +86-136-6760-0927

**Abstract:** This study presents an in-depth investigation into optimizing the mix design of ultra-high-performance concrete (UHPC) for enhanced sulfate erosion resistance, utilizing the modified Andreasen and Andersen (MAA) method. By testing the mechanical properties and slump flow of UHPC, it was determined that the optimal W/B = 0.2, and the best volume content of steel fibers is 2%. Through long-term tests lasting 360 days on three groups of UHPC specimens under different curing conditions, their mass loss, compressive strength corrosion resistance coefficient, surface appearance, and erosion layer thickness were tested. The results indicate that under sulfate attack, the mass and compressive strength corrosion resistance coefficients of UHPC specimens showed a trend of first increasing and then decreasing, due to the formation and expansion of ettringite and gypsum. The thickness of the erosion layer increases over time. By 360 days, the internal damage caused by sulfate attack is about twice as severe as it was after 60 days. However, the addition of steel fibers was found to effectively mitigate these effects, reducing mass loss and preserving the structural integrity of UHPC.

**Keywords:** ultra-high-performance concrete (UHPC); modified Andreasen and Andersen (MAA); sulfate erosion; mechanical property; microstructural

## 1. Introduction

In civil engineering, the stability and durability of concrete structures are adversely affected by a multitude of factors, including environmental degradation and chemical erosion, which result in a reduced service life and the introduction of potential safety hazards [1]. Sulfate attack, in particular, is identified as a critical factor compromising the durability of such structures [2]. This phenomenon involves a sequence of chemical reactions between sulfates and hydration products, either from the surrounding environment or within the concrete itself, culminating in the material's degradation [3]. Damage to concrete from external sulfate attack occurs through two primary mechanisms. The first is a chemical reaction wherein sulfates engage with cementitious materials to form ettringite and/or gypsum, leading to expansion. The second mechanism, cracking, arises from the interplay of chemical reactions and internal forces, attributed to the high permeability of the concrete. This allows sulfates to infiltrate the concrete and react with hydration products, resulting in the formation of expansive compounds such as gypsum and ettringite (AFt), thereby accelerating the destruction of the structure [4,5].

Ultra-high-performance concrete (UHPC) represents a significant advancement in engineering materials, exhibiting enhanced workability, mechanical properties, and durability compared to conventional concrete [6–9]. Characterized by its high-density structure, UHPC effectively impedes the penetration of corrosive agents [10–12]. Research conducted by Yang [13] examined the deterioration process of lightweight ultra-high-performance concrete (LUHPC) under flexural stress, exploring the damage mechanisms in UHPC specimens subjected to the combined effects of sustained flexural loading and sulfate attack. An investigation by Shannag [14] revealed that UHPC, when infused with 15% volcanic ash and 15% silica powder, retained over 65% of its strength after a year-long exposure to sulfuric acid, underscoring the material's robust resistance to sulfate corrosion. This resilience is attributed to enhanced pore refinement and densification within the transition zone of the specimens. Mbesa and Pera [15] assessed the ammonium sulfate resistance of UHPC with additives such as metakaolin, silica fume, and ground granulated slag, finding that specimens with silica fume exhibited the least mass loss. Further studies by Lee [16], Mangat, and El-Khatib et al. [17] corroborated the exceptional corrosion resistance of silica fume-doped concrete in sodium sulfate environments. However, it has been observed that an increase in silica fume and rice husk ash admixtures in concrete elevates strength loss in magnesium sulfate conditions [18]. The underlying cause is the depletion of silicates due to the volcanic ash admixture, prompting a direct interaction between $Mg^{2+}$ and calcium silicate hydrate (C–S–H) gels. This reaction transforms the C–S–H into non-bonding hydrated magnesium silicate (M–S–H) gels [19].

The design of the UHPC mix ratio aims to achieve an enhanced particle packing density, which is instrumental in decreasing porosity, bolstering strength, and augmenting impermeability. Predominantly, the Anderson–Andreasen model is utilized to determine the optimal mix ratio [20–22]. In 1907, Fuller and Thomson [23] introduced the "Fuller curve", delineating a maximum density grading curve for particle packing. Their exploration into continuous particle grading demonstrated that system porosity reaches its nadir when the particle size distribution curve for the aggregates in concrete is at saturation. Building on this foundation, Andreasen and Andersen refined the Fuller curve through the development of the modified Andreasen–Andersen Model (MAA) [24]. More recently, the application of response surface methodology (RSM) has facilitated further enhancements of MAA models [25,26]. This approach incorporates the D-optimization design method [27], which comprehensively addresses the interplay between numerous factors and the constraints existing among them. This method paved the way for the advanced quadratic saturated D-optimization design [28,29], recognized for its efficiency and statistical robustness. Importantly, this design approach is versatile, not constrained by the feedstock's state, and enables multifactorial studies under the most limited set of conditions. Ghafari [30] has proposed a statistical mixing design (SMD) method specifically for UHPC mixes. Employing the D-optimal design method, Wang et al. [31] tested the MAA model, selecting filling density as the pivotal response factor. These studies not only elucidate the intricate relationship among UHPC components but also reaffirm the high reliability of the MAA model, especially when the raw material system is comparatively straightforward.

The optimization of the UHPC mix ratio fundamentally hinges on achieving the highest densification of each raw material component. This is accomplished by calibrating the particle size distribution of each component to align with the principle of maximal packing density [32]. Research conducted by Soliman [33] has demonstrated that an increase in the matrix density correlates with enhanced compressive strength in UHPC. Consequently, the challenge of refining the UHPC mix ratio to simultaneously diminish carbon emissions and reduce economic costs emerges as a paramount concern [34]. In response to this challenge, Zhou [35] developed a three-dimensional discrete unit model for an eco-friendly high-performance concrete matrix at the micro-mechanical scale, thereby introducing a more scientifically grounded and effective mix design theory.

Despite the superior mechanical properties and durability of UHPC compared to traditional concrete, research has mainly focused on its mechanical properties, with less study

on its long-term durability, especially against prolonged sulfate attack. In light of this, this study utilized the principle of maximum particle packing density, combined with the modified Andreassen–Andersen (MAA) model, to formulate the matrix composition of UHPC. Then, the water-to-cement ratio and fiber content were adjusted based on workability and mechanical properties to determine the optimal UHPC mix design. The investigation lasted 360 days under two different treatment conditions: standard curing and sulfate attack, each cycle lasting 60 days. The evaluation of UHPC's long-term resistance to sulfate attack was conducted through a comprehensive analysis, including measurements of mass loss, determination of corrosion resistance coefficients for compressive strength, surface morphology examination, scanning electron microscope (SEM) imaging, and the use of ultrasonic nondestructive testing to assess the thickness of the damage layer. By observing the trends in mechanical properties and changes in both macroscopic and microscopic morphology, the study provides an intuitive understanding of UHPC specimens' resistance to sulfate attack over a long period and further quantifies the sulfate attack process through the thickness of the corrosion layer.

## 2. Materials and Methods

### 2.1. Materials

Portland cement P.O. 52.5 and Class I silica fume were selected for the study, with quartz powder serving as the cementitious material. Portland cement is produced by Shandong Fuzhu Building Materials Co., Ltd. in Shandong, China, while silica fume, quartz sand, and quartz powder are all produced by Henan Wuhu Environmental Protection Science and Technology Co. in Henan, China. The chemical composition of these materials is detailed in Table 1. Quartz sand was chosen as the fine aggregate, and the physical properties and particle size distribution of these granular materials are presented in Table 2 and Figure 1. To enhance the compatibility of the UHPC matrix, a polycarboxylic acid-based high-efficiency water-reducing agent was utilized. This agent, possessing a solid content of 17.0%, was added at a dosage of 1.0%, achieving a water reduction rate of 31%. The UHPC was further reinforced with steel fibers, each measuring 16 mm in length and 0.2 mm in diameter. The fibers under investigation are flat, copper-plated steel fibers, procured from Hongri Tektronix New Material Technology Co. (Zhejiang, China). A detailed overview of the principal properties of these steel fibers is presented in Table 3. The sulfate exposure tests were conducted using a solution prepared from anhydrous sodium sulfate and water, mixed to form a 10% concentration sodium sulfate solution.

**Table 1.** Chemical composition of cementitious materials used.

| Material | Chemical Composition (%) | | | | | | | | |
|---|---|---|---|---|---|---|---|---|---|
| | $SiO_2$ | $Fe_2O_3$ | $Al_2O_3$ | $CaO$ | $MgO$ | $K_2O$ | $Na_2O$ | $SO_3$ | Other |
| Cement | 20.30 | 4.12 | 4.91 | 64.8 | 1.05 | 0.51 | 0.16 | 1.75 | 2.4 |
| SF | 95.00 | 0.13 | 0.37 | 0.49 | 0.31 | 0.47 | 0.09 | 0.91 | 2.23 |
| QP | 99.13 | 0.21 | 0.19 | 0.13 | 0.08 | 0.21 | 0.01 | — | 0.04 |

**Table 2.** Physical properties and particle size distribution of the materials used.

| Material | Specific Density (g/cm$^{-3}$) | Specific Surface Area (m$^2$/g) | $D_{10}$ (μm) | $D_{50}$ (μm) | $D_{90}$ (μm) |
|---|---|---|---|---|---|
| Cement | 3.20 | 1.070 | 3.67 | 15.92 | 44.88 |
| SF | 2.71 | 0.128 | 23.21 | 92.40 | 284.40 |
| QP | 2.66 | 1.230 | 1.69 | 15.65 | 54.71 |
| QS | 2.55 | 0.009 | 461.37 | 628.74 | 858.88 |

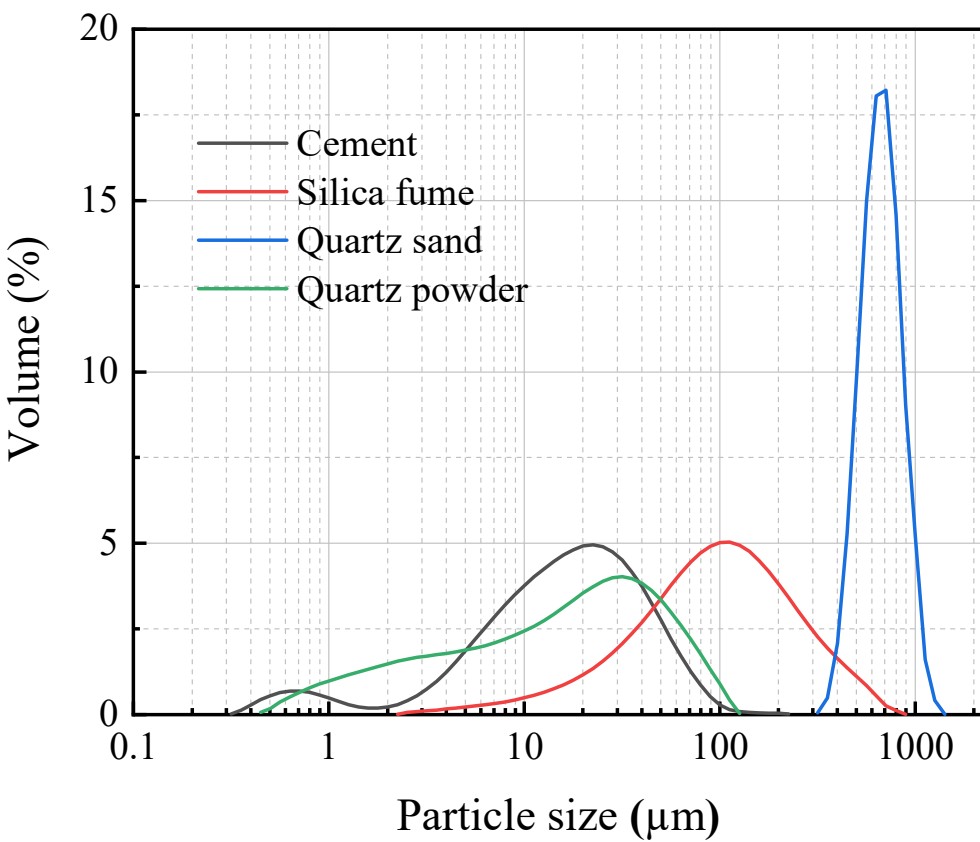

**Figure 1.** Particle size distribution of materials used.

**Table 3.** Physical properties of steel fibers used.

| Lengths (mm) | Diameter (mm) | Aspect Ratio (mm) | Tensile Strength (MPa) | Modulus of Elasticity (GPa) | Density (g/cm³) |
|---|---|---|---|---|---|
| 16 | 0.2 | 80 | 2870 | 230 | 7.8 |

*2.2. Experimental Methods*

2.2.1. Preparation of the UHPC

The raw materials required for the experiment were measured according to the predetermined design ratio. Subsequently, the mixture of water and the water-reducing agent was prepared, its total weight divided equally into two portions. Initially, the measured materials were introduced into the mixer and subjected to dry mixing for 30 s. During this process, half of the steel fibers were gradually added until uniform distribution was achieved. Following this, the first portion of the water and water-reducing agent mixture was incorporated into the mixture. Mixing then proceeded at a low speed for 120 s, after which the remaining half of the steel fibers was introduced. After a 30 s pause in mixing at low speed, the mixer's blades and the walls of the pot were scraped using a scraper to reintegrate any adhering slurry, coinciding with the addition of the second portion of water. Mixing continued for an additional 120 s or until the UHPC matrix exhibited a flowable consistency. The mixing method of the UHPC is shown in Figure 2.

Pour the fresh UHPC into molds, which are sized according to testing standards, and scrape off any excess mix from the top. After curing for 1 to 2 days at 20 ± 5 °C and more than 50% relative humidity, demold the samples. Then, cure the samples for 28 days in a standard curing chamber at 20 ± 2 °C and over 95% relative humidity.

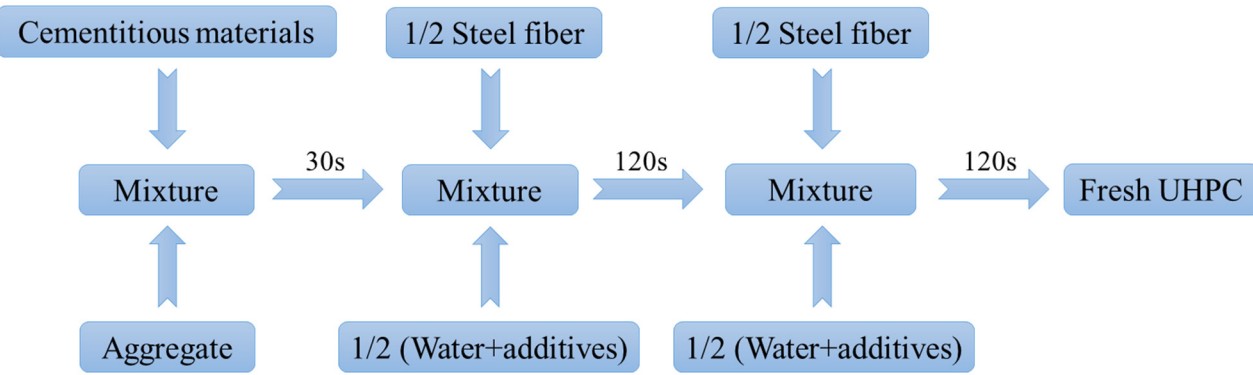

**Figure 2.** Mixing method of the UHPC.

### 2.2.2. Workability of the UHPC

The evaluation of the workability of UHPC was conducted through the measurement of its slump flow, in accordance with the guidelines stipulated by the Chinese standard GB/T 50080-2016 [36]. This procedure involved recording the maximum diameter of the slump flow as well as the diameter perpendicular to the direction of the maximum diameter, 50 s following the dispersal of the fresh UHPC onto a steel plate. The slump flow of UHPC was determined by calculating the average of these two diameters.

### 2.2.3. Mechanical Properties of the UHPC

The assessment of the compressive and flexural strengths of UHPC was conducted in compliance with the Chinese national standard GB/T 50081-2019 [37]. Specimens measuring 100 mm × 100 mm × 100 mm were utilized for the compression tests, while those for the flexural tests measured 100 mm × 100 mm × 400 mm. The compression tests proceeded at a loading speed of 0.8 MPa/s until failure of the specimen occurred. Compressive strength was determined for each specimen, with the mean value derived from three trials being designated as the compressive strength of the UHPC. Similarly, the flexural strength was evaluated using a four-point bending test at a loading rate of 0.08 MPa/s, where three parallel specimens for each UHPC batch underwent bending load tests. The flexural strength of the UHPC was then recorded as the average of the calculated values from these tests.

### 2.2.4. Sulfate Attack Resistance of the UHPC

Characterization procedures adhered to the Chinese national standard GB/T 50082-2009 [38]. The wet/dry cycle regimen encompassed a 24 h cycle, consisting of a 16 h immersion in a 10% sodium sulfate solution, ensuring a minimum liquid level of 20 mm above the specimen's surface. Subsequently, specimens were removed, air-dried for 1 h, and then oven-dried at 85 °C for 7 h, completing the cycle. Following a 28-day standard curing period, specimens were categorized into three groups for analysis: Group U1 comprised UHPC specimens without fiber, maintained under standard curing conditions; Group U2 included UHPC specimens without steel fiber, subjected to the wet/dry cycles for the duration necessary for the sulfate erosion test; and Group U3 consisted of UHPC specimens with a 2% steel fiber inclusion, also subjected to wet/dry cycles for the sulfate erosion test duration.

### 2.2.5. Erosion Layer Thickness and Apparent Morphology of the UHPC

The ultrasonic nondestructive evaluation was conducted on a prismatic specimen measuring 100 mm × 100 mm × 300 mm, subsequent to its exposure to sulfate erosion. A detection surface was designated on one face of the prism, measuring 100 mm × 300 mm, with the layout comprising five equidistant measurement points. The transmitting transducer was positioned at the detection surface's leftmost point and remained fixed, while the

receiving transducer sequentially moved from the second measurement point toward the right for detection. The detection process is shown in Figure 3, ultrasonic test equipment is shown in Figure 4. To enhance test data accuracy, both the transmitting and receiving transducers were coated with a coupling agent, minimizing friction between the transducer surfaces and the specimen's test surface. The assessment of internal damage within the concrete specimen was based on the test outcomes, utilizing a specified formula to calculate the variation in the thickness of the concrete's erosion layer across different erosion cycles. Retesting of specific measurement points was mandated in instances of abnormal data or wave amplitude during the evaluation. The thickness of the erosion layer was calculated employing Equations (1)–(3), grounded on the principle of identical ultrasonic wave wrapping and refraction times.

$$\frac{L}{V_b} = \frac{2\sqrt{H^2 + X^2}}{V_b} + \frac{L}{V_a} \tag{1}$$

$$H = \frac{L}{2}\sqrt{\frac{V_a - V_b}{V_a + V_b}} \tag{2}$$

$$L = \frac{A_1 V_a - A_2 V_b}{V_a - V_b} \tag{3}$$

where $H$ is the erosion layer of the concrete specimen (mm); $X$ is the projected length of the horizontal direction when the sound wave passes through the concrete (mm); $L$ is the distance between the two transducers at the inflection point of the speed of sound; $V_a$ is the speed of sound in the non-erosion layer (km/s); $V_b$ is the speed of sound in the erosion layer (km/s); $A_1$ is the intercept of the erosion layer (mm); and $A_2$ is the intercept of the non-erosion layer (mm).

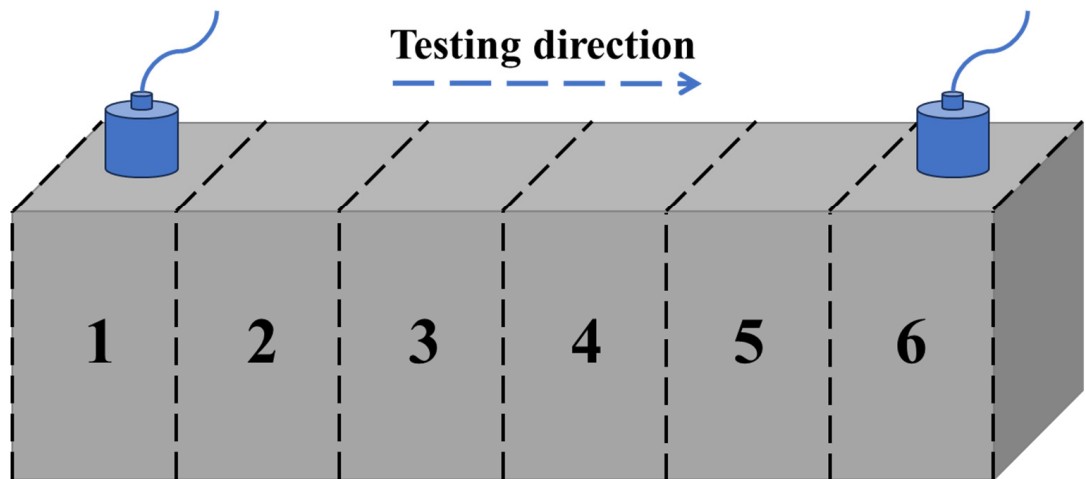

**Figure 3.** Schematic diagram of ultrasonic erosion test.

The measurement of the apparent porosity in UHPC specimens was conducted utilizing a photographic technique. A camera, equipped with a lens diameter of 30 mm, was employed to capture images across a surface area of 100 mm × 100 mm, followed by systematic captures at equal intervals on a 100 mm × 300 mm area. This methodology facilitated a detailed assessment of the porosity characteristics inherent in UHPC materials.

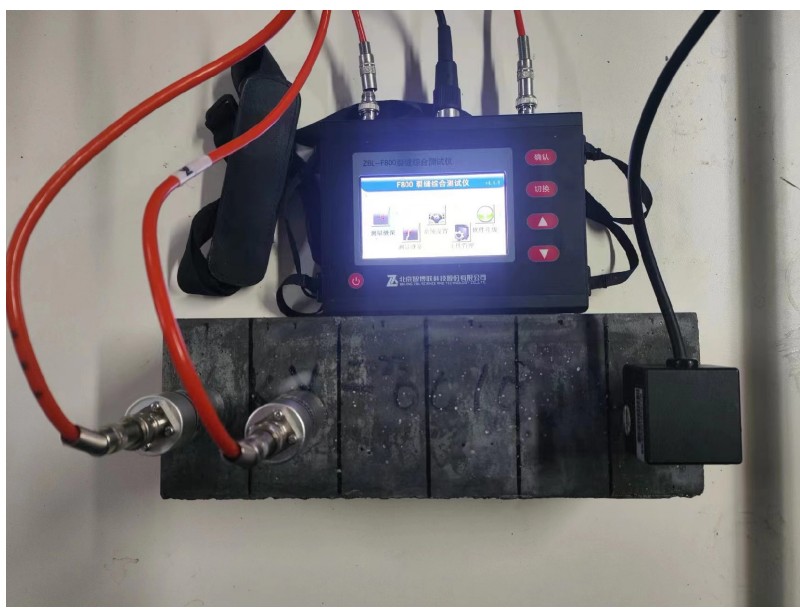

**Figure 4.** Ultrasonic test equipment.

2.2.6. Microscopic Morphology and Element of the UHPC

To meticulously examine the internal microstructure and elemental composition after 360 days of erosion, scanning electron microscopy (SEM, Gemini SEM 300, ZEISS, Oberkochen, Germany) and X-ray diffraction (XRD, Ultima IV, Rigaku, Tokyo, Japan) analyses were conducted on specimens from each group.

The micromorphology of UHPC specimens subjected to varying sulfate corrosion conditions was examined utilizing scanning electron microscopy (SEM). Specimen dimensions were approximately 3 mm by 3 mm, with an emphasis on minimal thickness and ensuring that surfaces were smooth, dry, and devoid of contaminants. These samples were subsequently placed in a blast drying oven set at 60 °C for a duration of 24 h, followed by sputter coating with gold to prepare them for analysis.

In a separate set of experiments, the UHPC specimens, representing different stages of exposure, underwent X-ray diffraction (XRD) testing. The gathered data were meticulously recorded and exported for further analysis. Subsequent interpretation of these results was conducted using Jade 6.5 software, facilitating a comprehensive understanding of the crystalline structure modifications induced by sulfate corrosion.

## 3. Design of the UHPC Based on MAA
### 3.1. Design of the UHPC Matrix

The modified Anderson model (MAA model, refer to Equation (4)) was employed for the proportion design of UHPC. Particle sizes for cement, silica fume, quartz powder, and quartz sand were determined using a laser particle size analyzer. Based on the derived particle size distribution, an optimal particle size distribution for granular materials was achieved under the principle of tightest packing density, resulting in a matrix characterized by elevated compactness.

$$P(D) = \frac{D^q - D_{min}^q}{D_{max}^q - D_{min}^q} \tag{4}$$

where $D$ is the particle size (μm); $P(D)$ indicates the proportion of solid particles smaller than the particle size D (%); $D_{max}$ is the maximum particle size; $D_{min}$ is the minimum particle size, and this test is 0.194 μm; $q$ denotes the distribution modulus, which is taken as 0.23 in this paper.

The adjustment of component proportions within the formulation was conducted utilizing MATLAB 2021b software, employing the least squares method (LSM) to align with the desired target curve, as depicted in Figure 5. This iterative process aimed at minimizing

the residual sum between the adjusted cumulative particle size curve and the target curve until compliance with the criteria set forth in the Chinese specification GB/T31387-2015 [39] was achieved. Specifically, this specification mandates that the silica fume content must constitute no less than 10% of the total cementitious materials, and the cement content should comprise at least 50% of the cementitious material volume. Upon meeting these requirements, the optimal quality ratio of the raw materials was established. The mixing ratio of the cementitious materials used is shown in Table 4.

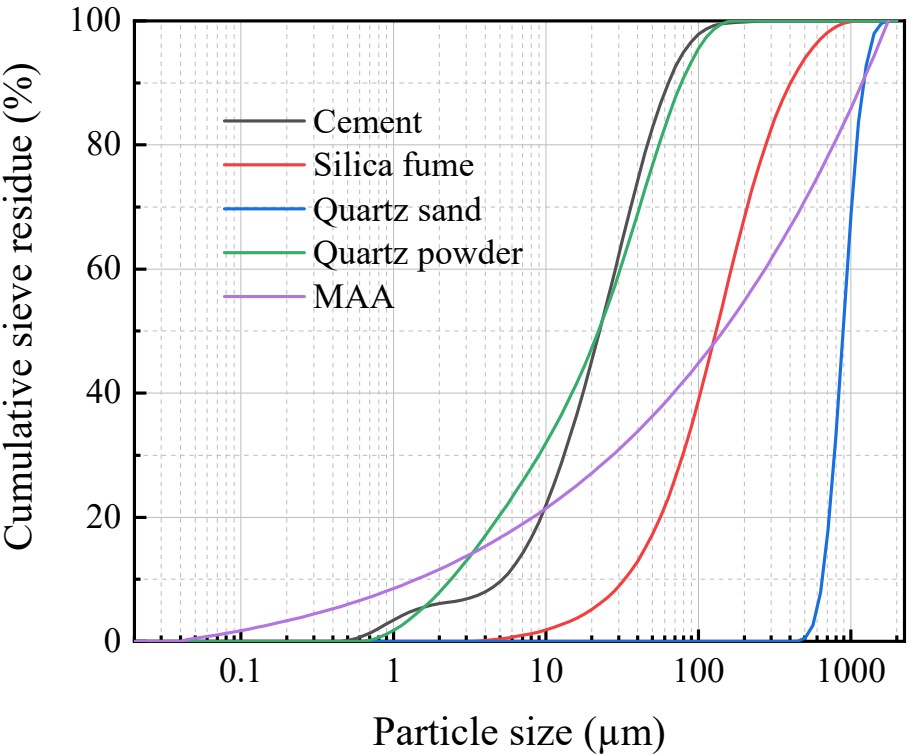

**Figure 5.** Particle size cumulative and target curves.

**Table 4.** Mixing ratio of the cementitious materials used.

| Material | Volume Fraction (%) | Mass Fraction (%) |
|---|---|---|
| Cement | 30.5 | 35.0 |
| SF | 9.0 | 9.0 |
| QP | 20.0 | 20.0 |
| QZ | 40.5 | 36.0 |

*3.2. W/B of the UHPC*

The modified Andreasen and Andersen approach (MAA model) is primarily focused on the inclusion of solid particulate materials within the raw material composition, yet it falls short of comprehensively addressing the entirety of the system's constituents. As indicated in Table 5, a water-to-binder ratio ranging from 0.18 to 0.22 was maintained for the UHPC matrix. This process involved the addition of both weighed aggregates and cementitious materials into a planetary mixer, where they underwent a dry mixing process for a duration of 30 s. Following this, a mixture of water and a water-reducing agent was introduced, facilitating continuous mixing for an additional 5 min. Subsequently, evaluations were conducted to ascertain the workability and mechanical properties of the concrete formulation. The mechanical properties of the UHPC are shown in Figure 6.

**Table 5.** Mixture proportions and the slump flow of the UHPC matrices.

| W/B | kg/m³ | | | | | | Slump Flow (mm) |
|---|---|---|---|---|---|---|---|
| | Cement | SF | QP | QS | Water | Water Reducer | |
| 0.18 | 716 | 187 | 416 | 743 | 237 | 13 | 196 |
| 0.19 | 716 | 187 | 416 | 743 | 251 | 13 | 217 |
| 0.20 | 716 | 187 | 416 | 743 | 264 | 13 | 227 |
| 0.21 | 716 | 187 | 416 | 743 | 277 | 13 | 235 |
| 0.22 | 716 | 187 | 416 | 743 | 290 | 13 | 227 |

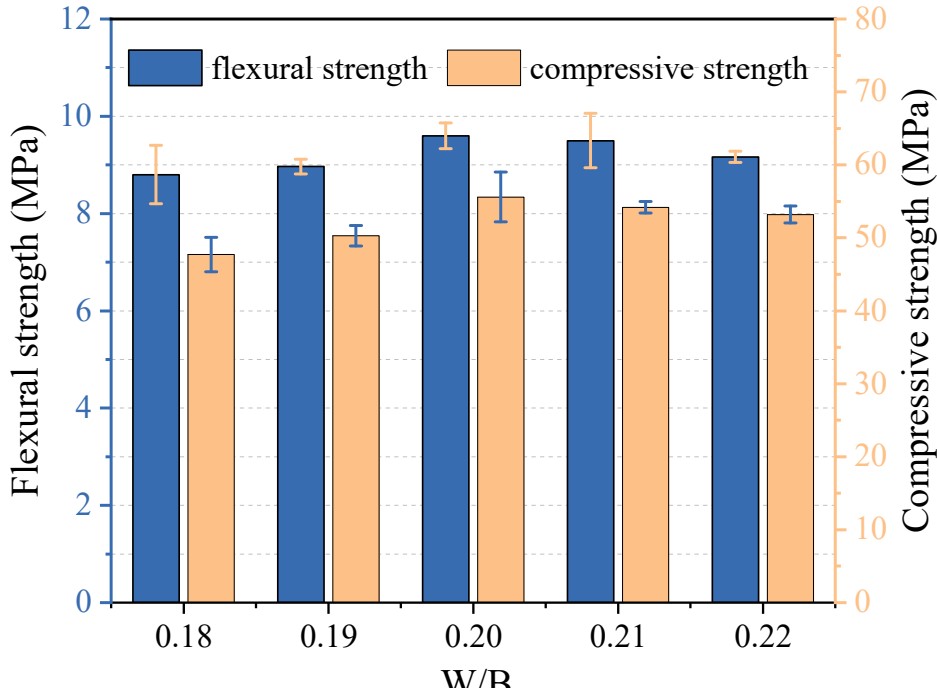

**Figure 6.** Three-dimensional flexural and compressive strength of UHPC.

The investigation reveals that the UHPC matrix exhibits enhanced workability and mechanical properties at a water-to-binder ratio (W/B) of 0.20. Analysis indicates a nuanced relationship between the W/B ratio and the UHPC's performance metrics: as the W/B ratio increases, both fluidity and flexural compressive strength initially rise, reaching a peak before subsequently declining. Notably, the pivotal point for mechanical properties is identified at a W/B ratio of 0.20, whereas optimal fluidity is observed at 0.21. This suggests that ratios below 0.20 are characterized by insufficient internal free water, leading to reduced fluidity and inadequate air expulsion. Conversely, ratios exceeding 0.20 may introduce excessive internal free water, potentially undermining the matrix's dense structure and diminishing its strength. Consequently, a W/B ratio of 0.20 is advocated within this study.

### 3.3. Steel Fiber Dosage of the UHPC

To investigate the effect of steel fiber reinforcement on UHPC, steel fibers were added to the mixtures in dosages of 0%, 1%, 1.5%, 2%, and 2.5% by volume. The resulting UHPC specimens were assessed for their degree of expansion, and their flexural and compressive strengths at various ages. Figures 7 and 8 present the findings, which were instrumental in identifying the fiber dosage that exhibited enhanced performance for subsequent research phases.

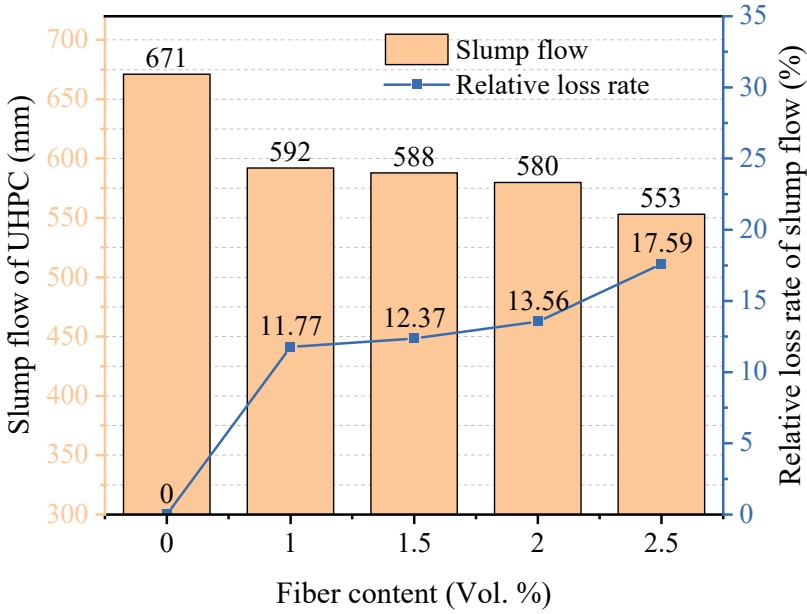

**Figure 7.** Slump flow of UHPC with different fiber dosages.

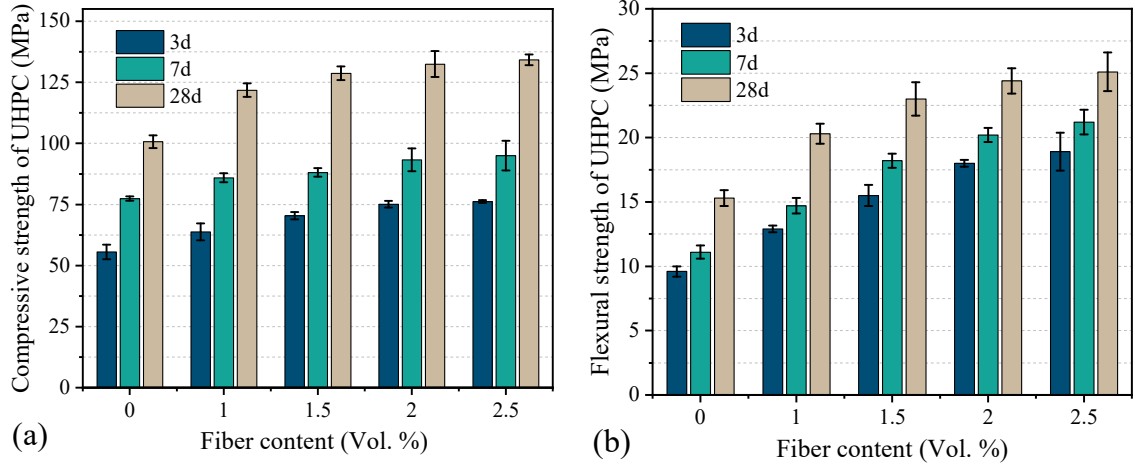

**Figure 8.** Strength of UHPC with various fiber volume fractions at different ages: (**a**) flexural strength of UHPC; (**b**) compressive strength of UHPC.

Observations indicate a decline in the expansion rate of UHPC as the volume fraction of steel fibers increases. Compressive and flexural strengths of specimens, assessed at 28 days and across different fiber volume fractions, satisfy the established specifications. The incorporation of steel fibers at a 2% volume fraction markedly improves the compressive and flexural strengths while preserving fluidity. Thus, a 2% steel fiber volume fraction is adopted for further studies on sulfate erosion resistance. Mixture proportions of the UHPC specimens resistant to sulfate attack are shown in Table 6.

**Table 6.** Mixture proportions of the UHPC specimens resistant to sulfate attack.

| Sample ID | kg/m³ | | | | | | Fiber Content (Vol. %) |
|---|---|---|---|---|---|---|---|
| | Cement | SF | QP | QS | Water | Water Reducer | |
| U1 | 716 | 187 | 416 | 743 | 264 | 13 | 0 |
| U2 | 716 | 187 | 416 | 743 | 264 | 13 | 0 |
| U3 | 716 | 187 | 416 | 743 | 264 | 13 | 2 |

## 4. Results and Discussion

*4.1. Sulfate Attack Resistance of the UHPC*

4.1.1. Relative Mass Loss

The change in relative mass loss of UHPC samples is shown in Figure 9. Analysis of the graph indicates a gradual increase in the mass of U1 group specimens over a 360-day curing period. Conversely, specimens from the other two groups exhibit a trend of initial mass gain followed by a loss, attributed to alternating wet/dry cycles and sulfate erosion effects. The mass increment in the early stages is likely due to ongoing hydration, facilitated by the specimens' water-to-binder ratio, a process extending beyond 360 days. Regarding sulfate erosion, the observed mass augmentation can be linked to ettringite formation, as opposed to gypsum formation, which predominates in high sulfate ion concentrations. The process entails an initial reaction of sulfates with concrete's calcium hydroxide to form calcium sulfate, which then reacts with calcium aluminates, leading to ettringite formation. Conversely, gypsum formation is triggered by the reaction of calcium hydroxide with sulfate ions, resulting in gypsum ($CaSO_4 \cdot 2H_2O$).

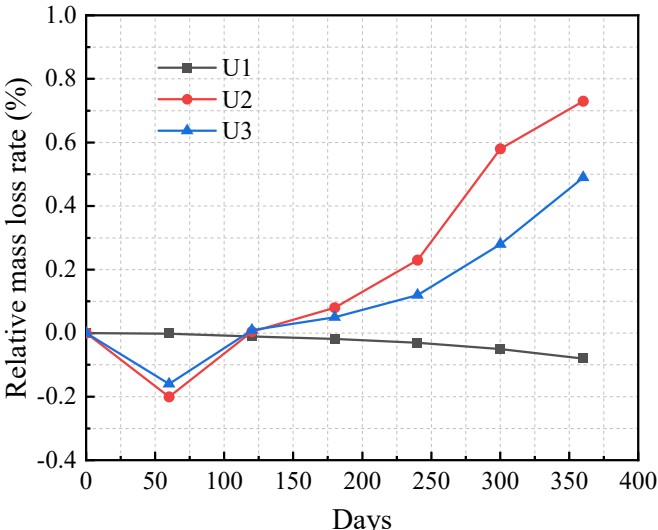

**Figure 9.** Relative mass loss of different UHPC specimens: U1 represents UHPC specimens without fibers under standard curing conditions; U2 represents UHPC specimens without fibers under conditions of sulfate erosion; U3 represents UHPC specimens with steel fibers under conditions of sulfate erosion.

The formation of water-insoluble gypsum and ettringite crystals in the described reactions initially leads to a densification of the concrete matrix's pore structure, thereby increasing the specimens' mass. However, as the reaction progresses, a gradual decrease in the specimens' mass is observed across all test groups, eventually falling below their initial mass after approximately 60 days. This trend may be attributed to the reaction of $SiO_2$ present in the raw materials with $Ca(OH)_2$, diminishing the likelihood of $Na_2SO_4$ interacting with $Ca(OH)_2$. Concurrently, the volumetric increase in gypsum, resulting from the erosion reaction, induces internal stresses by expanding the matrix. These stresses exacerbate pre-existing microcracks, leading to their further propagation. Once these cracks reach a critical size, spalling of the specimens occurs, resulting in a decline in specimen quality and an increase in relative mass loss.

With the increase in erosion duration, a higher volume fraction of fibers has been linked to lower relative mass loss, highlighting the effectiveness of steel fibers in counteracting the detrimental impacts of sulfate erosion on concrete. This effect is attributed primarily to the continuous hydration of the cement-based materials, which fortifies the fiber–matrix interface transition zone. Moreover, the expansion of gypsum and ettringite fills the concrete pores and the transition zone at the fiber–matrix interface, further enhancing this

effect. The culmination of these factors leads to a denser matrix structure, reduced rate of crack formation and growth, less spalling, and ultimately, a decrease in the relative mass loss of the specimens.

### 4.1.2. Compressive Strength Corrosion Resistance Coefficient of UHPC

Figure 10 illustrates the variations in the corrosion resistance coefficient of compressive strength across a range of ultra-high-performance concrete specimens. Under standard curing conditions at 360 days, the compressive strength and corrosion resistance coefficient of UHPC specimens exhibited a slight increase from their initial values, demonstrating a monotonic upward trend. However, specimens subjected to the most severe sulfate erosion maintained a corrosion resistance coefficient of approximately 0.9. The results indicate that, under the combined effects of sulfate exposure and wet/dry cycling, the compressive strength and corrosion resistance coefficient of the test specimens first increased and then decreased. In the early stages of erosion (first 60 days), a phase where the corrosion resistance coefficient exceeded 1 was observed, suggesting that the compressive strength of UHPC was superior to its initial strength during this period. This enhancement could be attributed to the early-stage erosion-induced chemical reactions, which resulted in the pore structure of the matrix being filled and compacted, thereby densifying the overall structural matrix. Groups U2 and U3 specimens' compressive strength and corrosion resistance coefficients rapidly reached their peak values during the initial phase of wet/dry cycling, followed by a monotonic decrease, falling below the initial compressive strength and corrosion resistance coefficient after approximately 60 days. This decline could be partly due to the expansion caused by ettringite, which absorbed a significant amount of water molecules, leading to its radial growth and the exertion of immense internal stress through mutual compression among ettringite crystals, ultimately causing damage to the concrete structure. Additionally, the erosion process may deplete calcium silicate hydrate (C–S–H), densely distributed between the matrix particles and significantly reducing the porosity within the matrix, thereby enhancing the strength of UHPC. The substantial consumption of C–S–H during erosion leads to a softening effect within the cementitious matrix, reducing its binding force and ultimately resulting in a decrease in specimen strength. The compression damage pattern of UHPC specimens under different conditions after 360 days is shown in Figure 11.

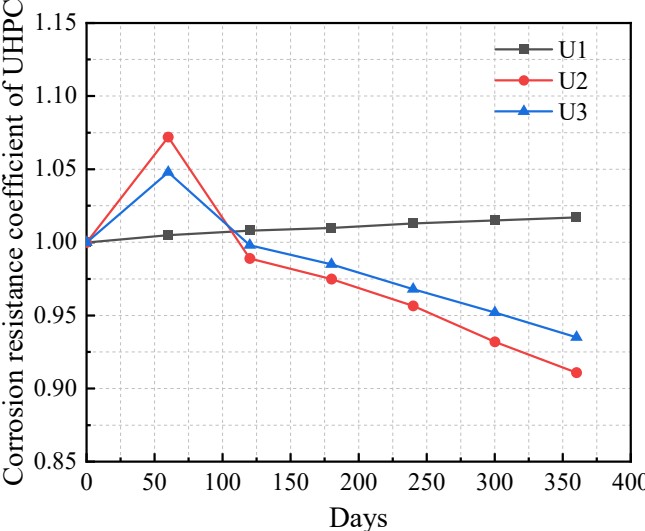

**Figure 10.** Corrosion resistance coefficients of compressive strength: U1 represents UHPC specimens without fibers under standard curing conditions; U2 represents UHPC specimens without fibers under conditions of sulfate erosion; U3 represents UHPC specimens with steel fibers under conditions of sulfate erosion.

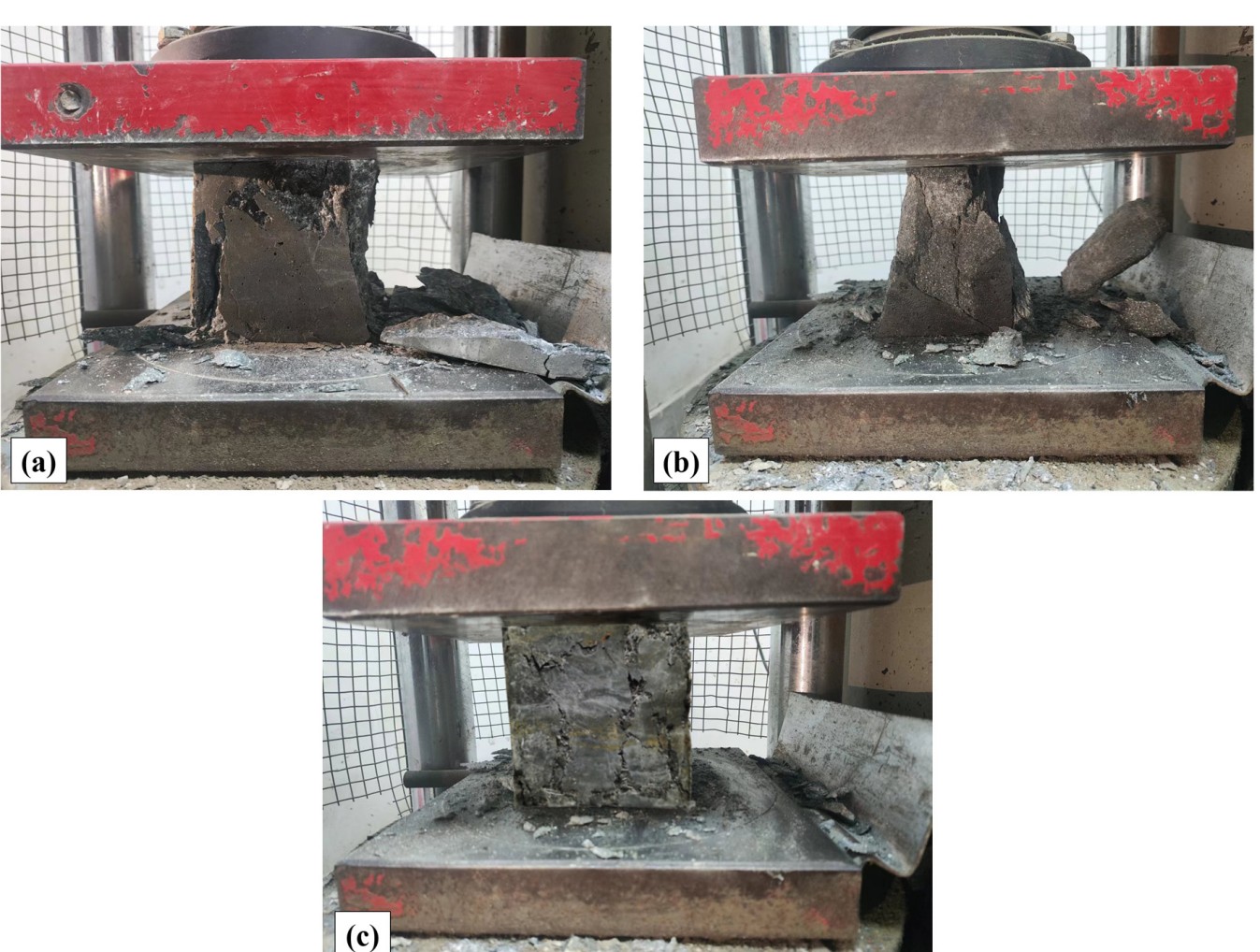

**Figure 11.** Strength compressive damage pattern under different conditions in 360 days: (**a**) U1; (**b**) U2; (**c**) U3.

As the erosion process progressed, the compressive strength corrosion resistance coefficient of UHPC specimens increased with the addition of fiber content, indicating that steel fibers play a role in enhancing the corrosion resistance coefficient of UHPC's compressive strength. This improvement could be attributed to the densification of the structure caused by the hydration of the cementitious materials and the expansion of gypsum and ettringite, as mentioned earlier. Additionally, the failure occurred at the interface transition zone between the steel fibers and the matrix (as shown in Figure 12b), where, even after failure, a high degree of integrity between the matrix and the steel fibers was maintained (as illustrated in Figure 12a). This suggests that the interface transition zone is a relatively weak part of the structure. Under standard curing conditions, UHPC specimens without fibers exhibited a more uniform failure mode, whereas specimens not containing fibers and subjected to sulfate erosion showed the greatest degree of damage and non-uniformity. Specimens with a 2% fiber content displayed the least degree of compressive failure after sulfate erosion, demonstrating that the addition of steel fibers can effectively maintain structural integrity. Damage locations around the specimens of all groups were characterized by flaky bulging.

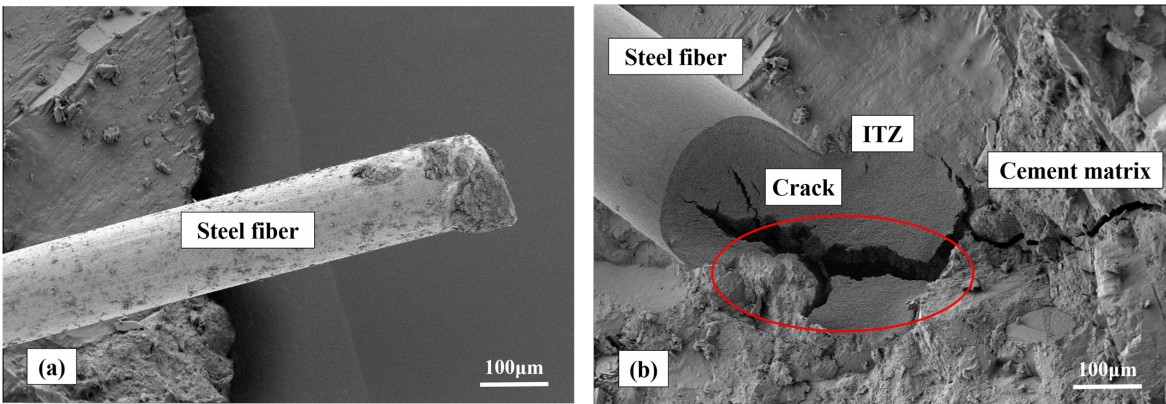

**Figure 12.** Steel fiber–matrix interface zone: (**a**) steel fiber after compression damage; (**b**) concrete interface transition zone damage.

### 4.1.3. Surface Morphology Analysis of UHPC

UHPC in different periodicities of wet/dry cycles and sulfate under the joint action of the surface morphology is shown in the following figures: Figure 13a illustrates a specimen after 28 days of standard curing, while Figure 13b represents a control group specimen subjected to a natural curing cycle of 360 days. A comparison of these specimens reveals that, over 360 days, specimens with larger apparent pores exhibit a more irregular state, whereas those with smaller pores remain relatively unchanged. This suggests that an increased water–gel ratio may permit minor new damage during the continued hydration of UHPC. Figure 13c,d display UHPC specimens after 360 days of wet/dry cycles, with Figure 13d showing signs of surface spalling and more extensive and deeper erosion holes than observed in Figure 13c. These changes indicate that the severity of sulfate–induced erosion in UHPC specimens escalates with prolonged wet/dry cycles, but the incorporation of steel fibers can mitigate the effects of sulfate erosion to some degree.

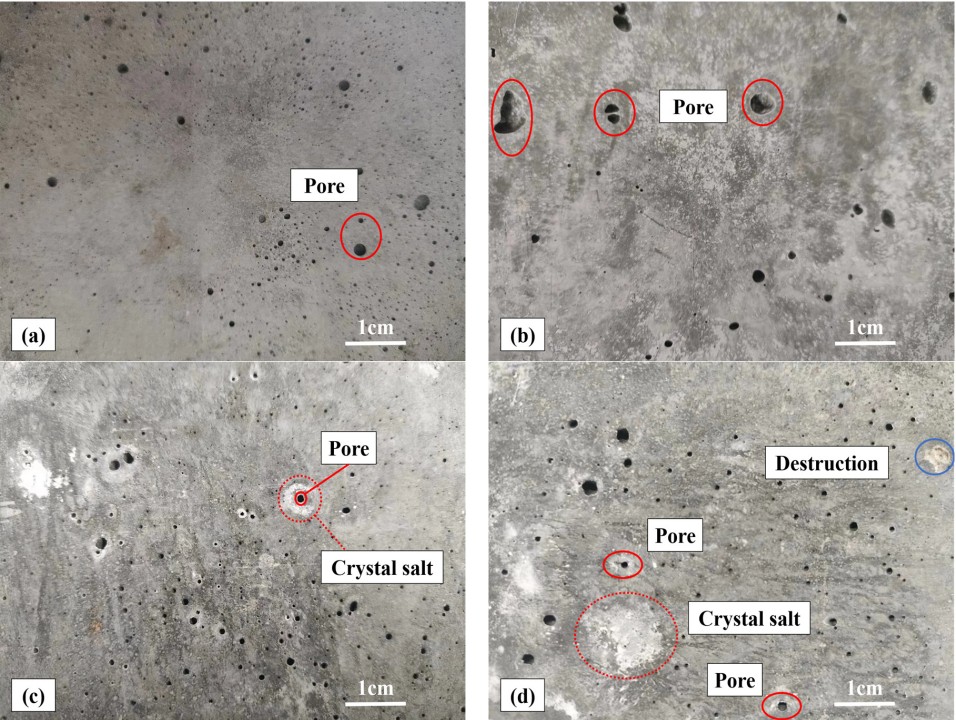

**Figure 13.** Surface of UHPC: (**a**) the specimen of 28 days standardized curing; (**b**) 360 days—U1; (**c**) 360 days—U3; (**d**) 360 days—U2.

Upon evaluating the effects of 360 days of corrosion, it was observed that within the same specimen, the size of the pores unaffected by crystallization closely approximated those affected by crystallization. Furthermore, the presence of crystalline substances precipitated around the pores led to a slight reduction in the size of pores influenced by crystalline salts compared to normal pores. A comparison between specimens without fibers and those with a 2% fiber content after 360 days of corrosion revealed that the visible pore sizes in both types of specimens ranged between 0.1 mm and 0.5 mm, categorizing them as harmful large pores that affect the structure. Under the same curing duration, the size of the visible pores on the specimens also fell within this range, indicating that the dense nature of UHPC typically results in pore sizes smaller than 0.5 mm, which do not further expand under 360 days of sulfate corrosion.

### 4.1.4. Erosion Layer Thickness Analysis

Upon analyzing the mass loss of ultra-high-performance concrete (UHPC) and the changes in the corrosion resistance coefficient of its compressive strength, it was noted that, in the first 60 days of exposure to sulfate corrosion, a notable phenomenon was observed: the densification of the C–S–H gel within the pores, leading to enhanced properties. The study includes ultrasonic testing of the erosion layer thickness in UHPC over a 60–day period, with the findings detailed in Figures 14–16.

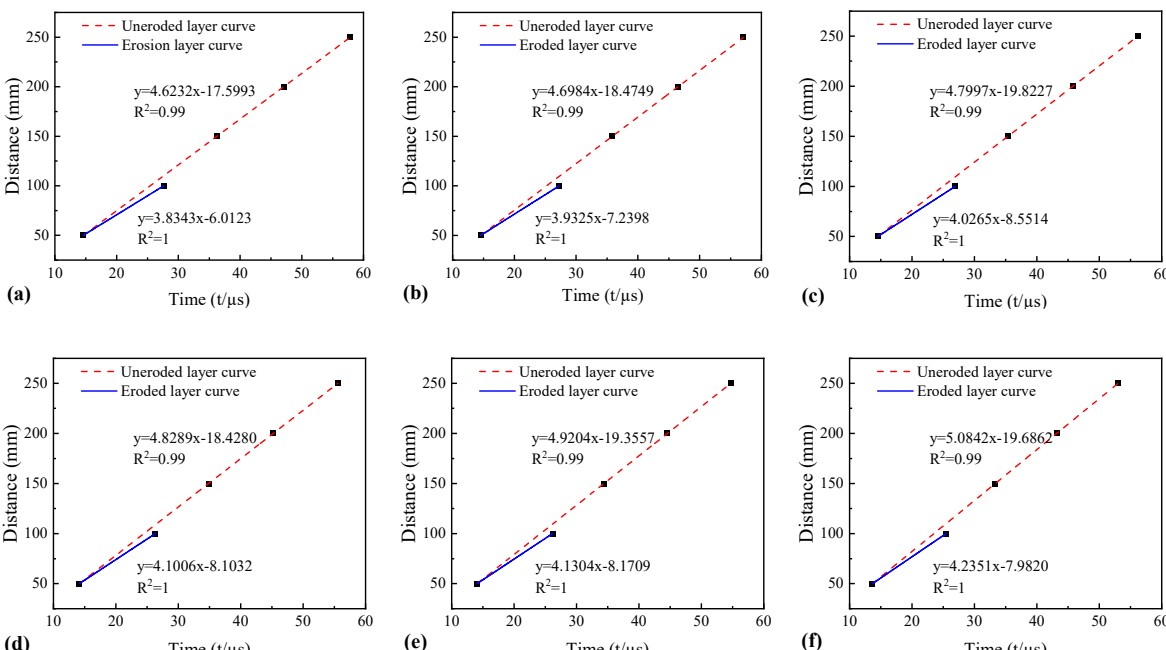

**Figure 14.** Time–distance diagram of U1: (**a**) 60 days; (**b**) 120 days; (**c**) 180 days; (**d**) 240 days; (**e**) 300 days; (**f**) 360 days.

The results indicate that due to the variation in sound speed between erosion and unerosion layers, the sound speed–distance relationship (i.e., the slope) also differs, with a lower slope representing the ultrasonic wave passing through the erosion layer, and a higher slope indicating passage through the unerosion layer, with both intersecting at a single point. Over time, the slope of the sound time–distance line for the erosion concrete layer showed an increasing trend, suggesting a reduction in the ultrasonic wave transmission path and a corresponding increase in sound speed. This phenomenon can be attributed to the early stages of sulfate wet–dry cycles, where the hydration products formed from the reaction between sulfate ions in the concrete and the external environment fill the internal pores of the concrete, rendering the specimen denser. Consequently, the path of the ultrasonic wave through the concrete shortens, leading to an increase in sound speed. However, as the cycle reaches 360 days, the slope of the sound time–distance for the

damaged layer gradually decreases, indicating an elongation in the path as microcracks emerge and develop within the concrete specimen. The sound speed–distance slope for the ultrasonic waves passing through the undamaged layer in groups U2 and U3 initially decreases and then increases, differing from the damaged layer. This suggests that the hydration processes in the two zones are not identical, and the wet–dry cycle has a distinct impact on hydration at different locations.

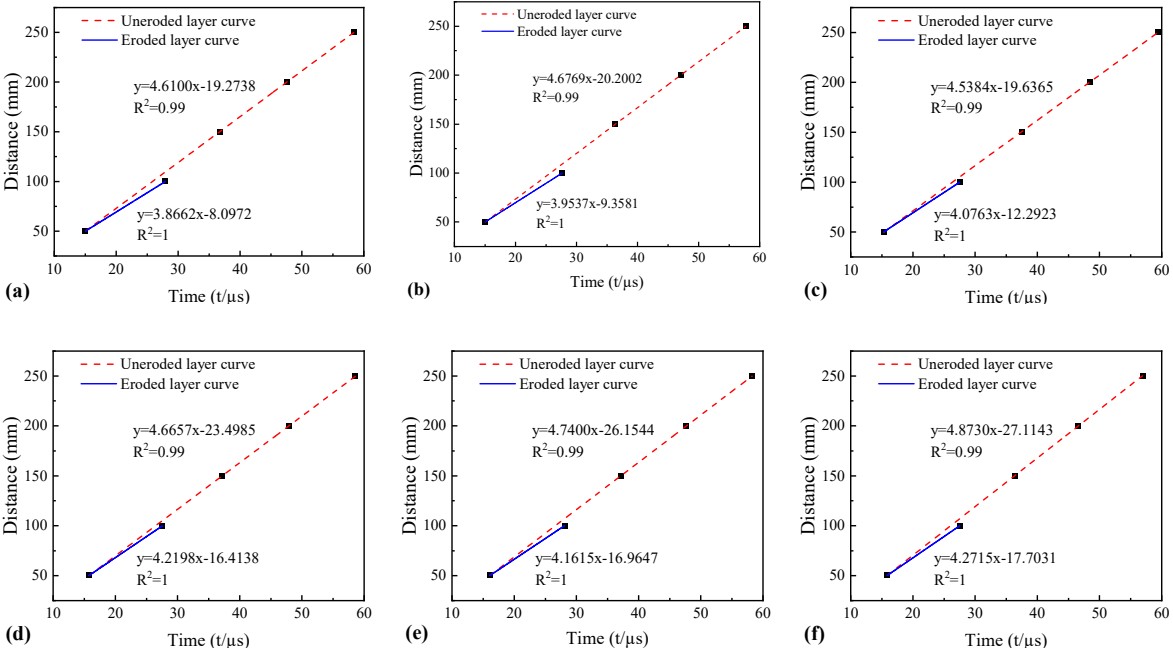

**Figure 15.** Time–distance diagram of U2: (**a**) 60 days; (**b**) 120 days; (**c**) 180 days; (**d**) 240 days; (**e**) 300 days; (**f**) 360 days.

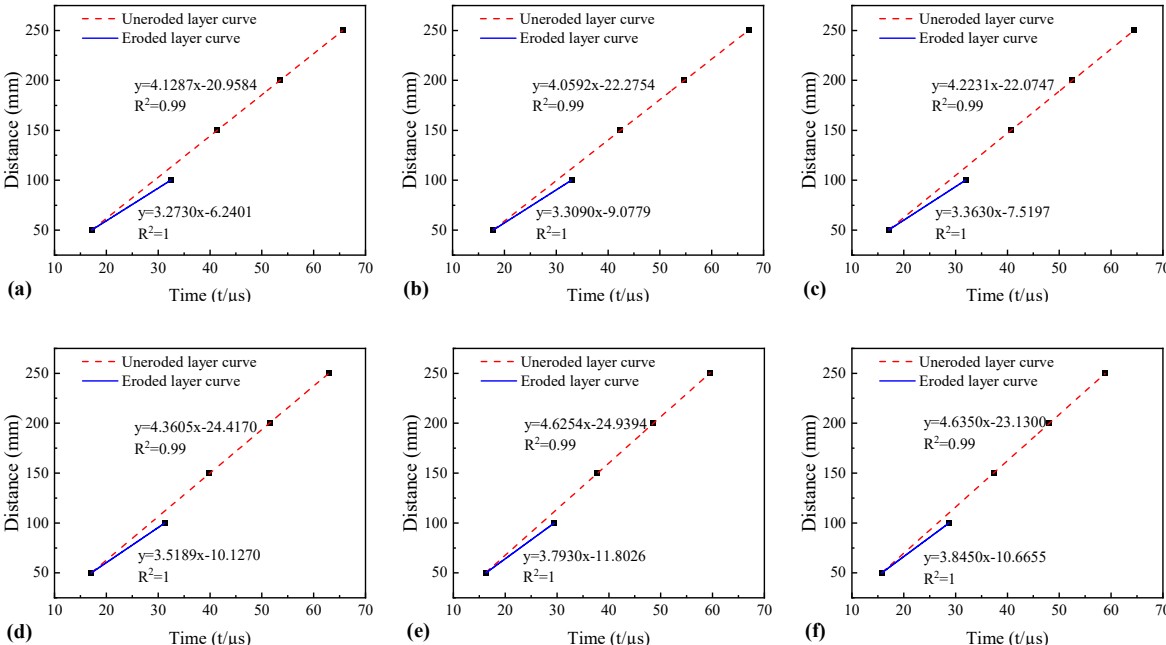

**Figure 16.** Time–distance diagram of U3: (**a**) 60 days; (**b**) 120 days; (**c**) 180 days; (**d**) 240 days; (**e**) 300 days; (**f**) 360 days.

The thickness of the erosion layer of UHPC is shown in Figure 17. The study's findings indicate an upward trend in the thickness of the erosion layer as sulfate corrosion progresses over time. Specifically, the erosion layer thickness for the U1 group exhibited a 35.4% increase, the U2 group a 122.4% increase, and the U3 group a 67.5% increase. This trend underscores that prolonged exposure to a moist environment, even under standard curing conditions, inevitably leads to external-to-internal specimen erosion. Upon exposure to sulfate erosion, the increase in erosion layer thickness surpasses that observed under standard conditions for equivalent durations. Moreover, the rate of increase in erosion layer thickness due to sulfate exposure accelerates over time, with the thickness doubling after 360 days. This observation confirms the effectiveness of sulfate erosion against the concrete's dense structure. However, specimens reinforced with 2% steel fibers, despite having a thicker erosion layer than the U1 group under standard conditions, showed a reduced thickness growth rate at 360 days compared to the U2 group. This suggests that the incorporation of steel fibers into the concrete matrix enhances its density, thereby effectively mitigating the impact of sulfate erosion. The schematic diagram of the sulfate erosion process in UHPC is shown in Figure 18.

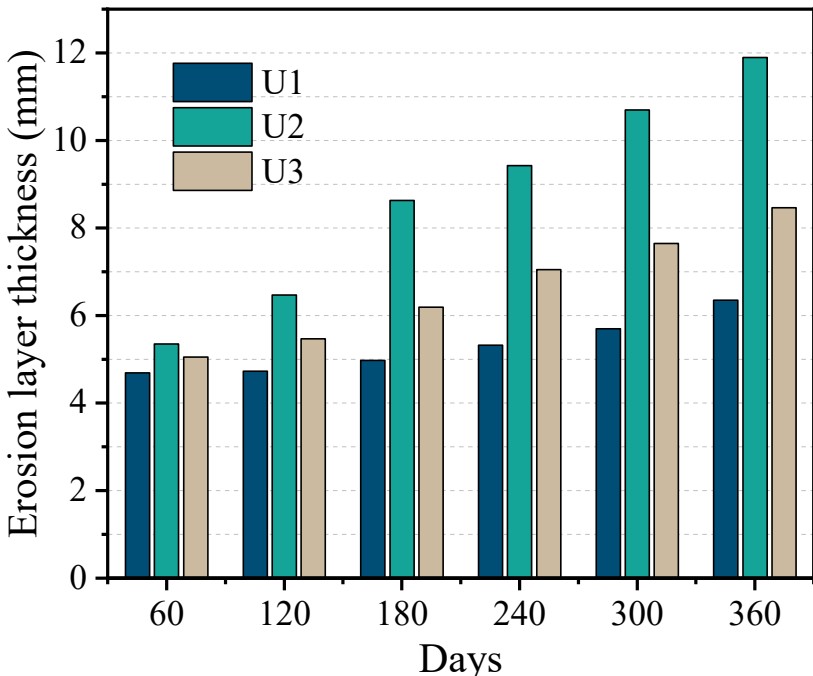

**Figure 17.** The thickness of the erosion layer of UHPC.

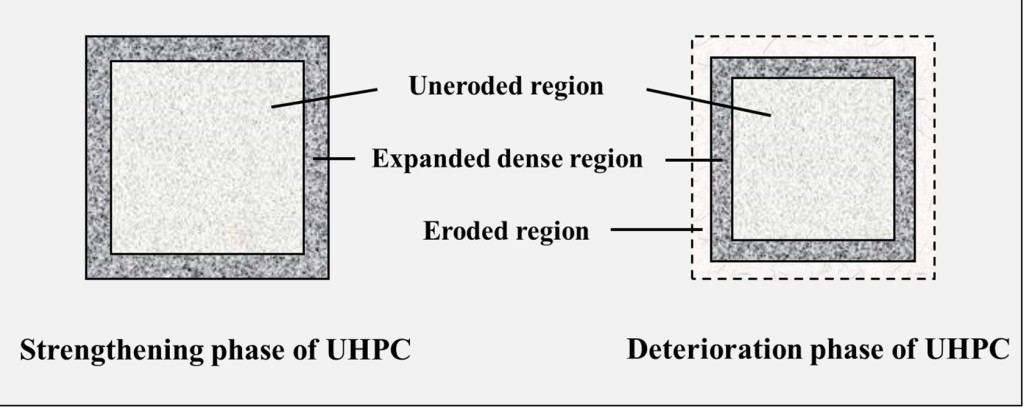

**Figure 18.** Sulfate erosion process in UHPC.

### 4.2. Microstructure Analysis of UHPC

#### 4.2.1. SEM

Figures 19a, 20a, 21a and 22a illustrate the presence of pore structures and microcracks across all specimen groups. It is identified that pores exceeding 100 nm in diameter within the cementitious matrix are deemed hazardous. These pores significantly compromise the mechanical integrity and durability of the material, potentially leading to an expedited deterioration of the structure [40]. Microscopic examination of each specimen group reveals the presence of discernible cloudy or flocculent material, identified as calcium silicate hydrate (C–S–H), alongside soft prismatic needles indicative of ettringite (AFt). This can be seen in Figures 19b, 20c, 21b and 22d. These constituents are recognized as critical by-products of the hydration process. The substantial presence of C–S–H across the specimens serves as evidence that the hydration process of UHPC reaches completion after a period of 360 days, it can be seen in Figures 19b and 20c. Furthermore, the formation of AFt is observed to impede the progression of the hydration reaction. The cement hydration products undergo consolidation, forming in lamellar clusters, which accounts for the observed laminar expansion along the flanks as the concrete fractures through the containment ring, as depicted in Figure 21c. Within the non-corrosive zones of groups U2 and U3, cement particles were detected, with Figure 21d showcasing particles ranging in size from 1 to 50 µm. This observation suggests the potential for additional hydration in UHPC beyond the 360-day benchmark under standardized conditions.

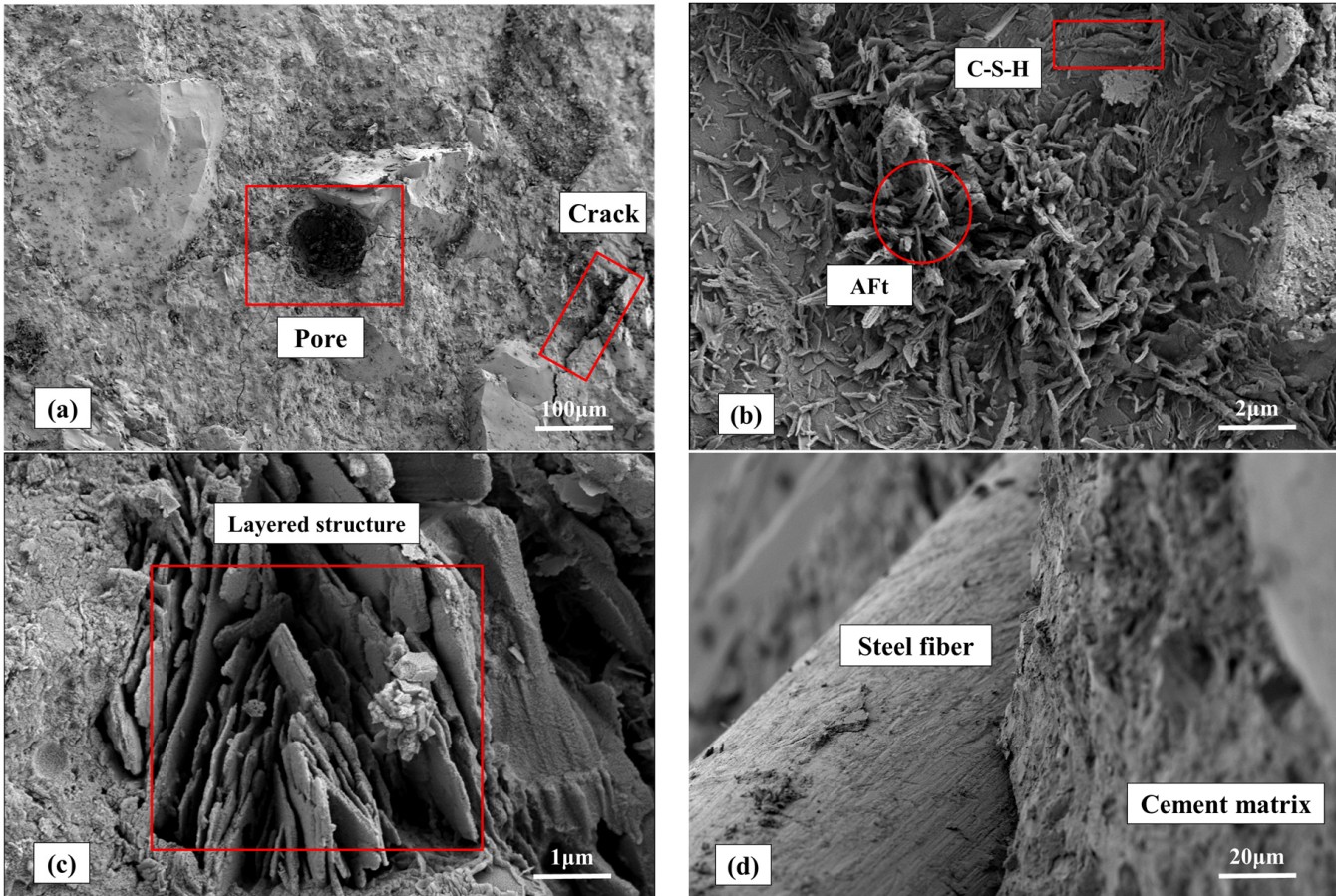

**Figure 19.** Erosion layer of U3: (**a**) pores and cracks; (**b**) AFt and C–S–H; (**c**) layered structure; (**d**) steel fiber.

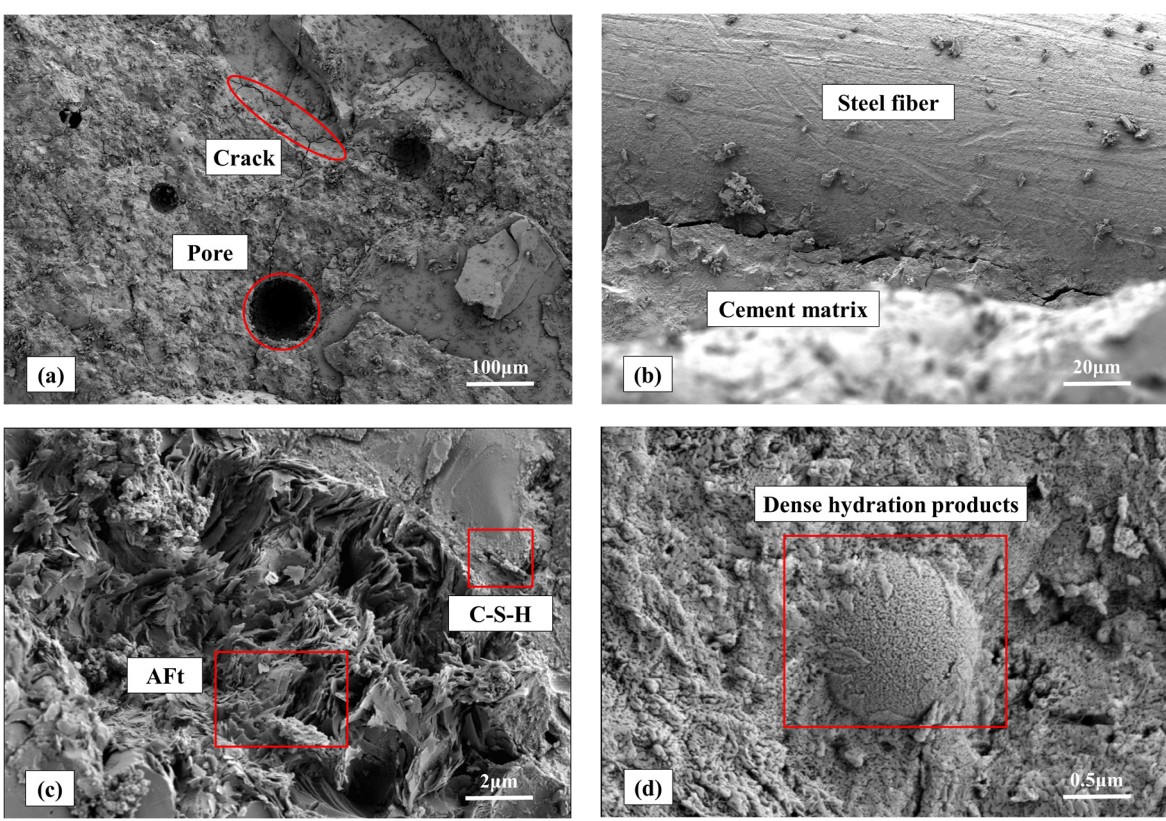

**Figure 20.** Non-erosion layer of U3: (**a**) pores and cracks; (**b**) steel fiber; (**c**) AFt and C–S–H; (**d**) dense hydration products.

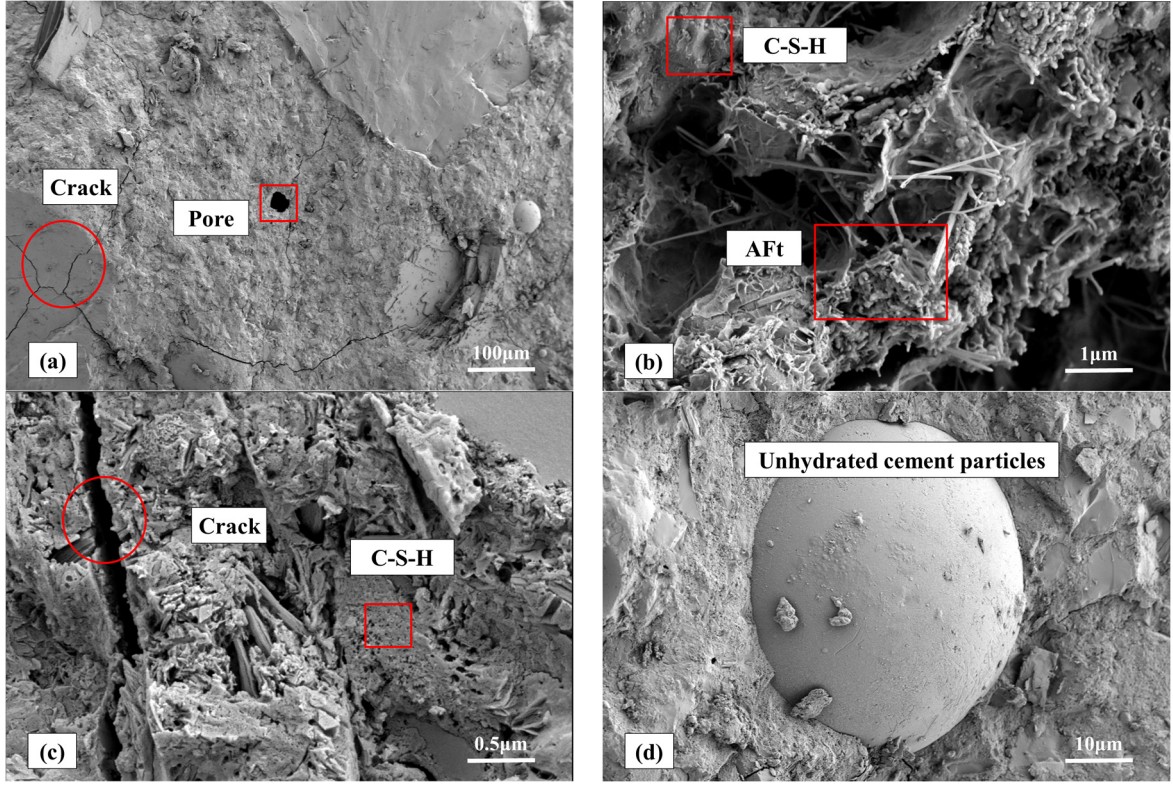

**Figure 21.** Erosion layer of U1: (**a**) pores and cracks; (**b**) AFt and C–S–H; (**c**) C–S–H and cracks; (**d**) un-hydrated cement particles.

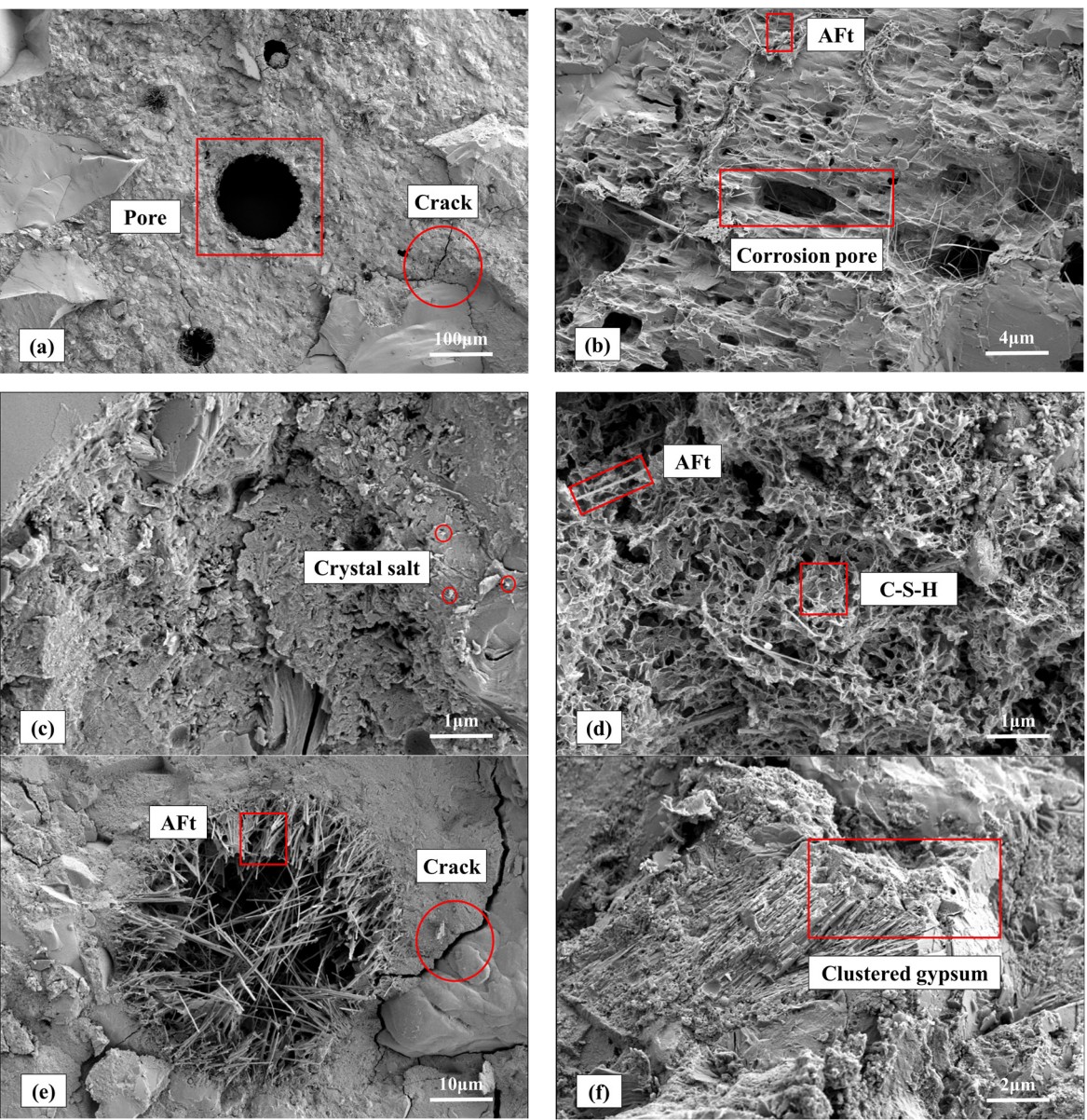

**Figure 22.** Erosion layer of U2: (**a**) pores and cracks; (**b**) corrosion pore; (**c**) crystal salt (**d**) C–S–H and AFt; (**e**) AFt and cracks; (**f**) clustered gypsum.

As illustrated in Figure 20d, there is a noted reduction in the size of the large particles during the latter stages of hydration, constrained by the limited growth space between particles. This limitation culminates in the deposition of hydration products on the surfaces of the particles, indicating a transition in the hydration dynamics as the process matures. The emergence of clinker particles results in the formation of a protective layer enveloping these particles [41]. However, within the erosion zones of groups U2 and U3, these clinker particles become less discernible, indicating that the wet/dry cycling process not only induces erosive damage but also facilitates a more comprehensive hydration process. Comparative analysis between Figures 19d and 20b indicates a discernibly rougher texture on steel fibers within the erosion zone compared to those in non-erosion areas, highlighting enhanced erosion. This research demonstrates that pore pressure evolution in UHPC specimens under stress deviates from conventional concrete behavior, significantly influenced by fiber content. The incorporation of fibers is proven to bolster structural stability. In instances where the fiber volume fraction is zero, an emergence of corroded pores with

an irregular distribution and dispersed crystalline salts is evident, overlaying original pore structures.

Notably, Figure 22e reveals a coexistence of ettringite (AFt) and calcium silicate hydrate (C–S–H) gel within these pores, capable of mitigating microcrack formation and inhibiting crack propagation resulting from drying or chemical shrinkage during cement hydration [42,43]. Figure 22f illustrates the presence of clustered gypsum formation within the specimen, attributed to an abundance of sulfate ions in the immersion solution reacting with calcium ions to produce gypsum. This observation signifies a heightened degree of corrosion in this zone, with more pronounced corrosion phenomena compared to specimens with a 2% steel fiber doping rate. This finding further indicates that UHPC reinforced with steel fibers exhibits enhanced resistance to sulfate attack, with the fiber-reinforced skeleton structure offering superior protection against such degradation.

### 4.2.2. XRD

The XRD test spectra, as depicted in Figure 23a, demonstrate a notable similarity across all specimen groups at 360 days, with group U1 exhibiting the most pronounced and regular wave peaks. This observation suggests that the UHPC specimens in group U1 are both intact and have undergone sufficient hydration. Conversely, group U2 displays significantly lower peak values compared to other groups, indicating a reduction in strength and crystallization quality following sulfate attack, marking it as the most severely eroded group. The comparison between erosional and non-erosional zones within group U3 reveals minor differences in peak values, implying a similar degree of hydration, albeit with non-erosional zones showing marginally higher peaks, suggesting a slight impact of erosion on the overall structure. Figure 23b highlights $SiO_2$ as the predominant component in the U1 group, attributed to the substantial presence of silica fume, quartz sand, and quartz powder in the raw materials, which maintain their abundance post-hydration. Furthermore, the presence of significant post-hydration products at 360 days indicates a more complete hydration process and a denser structure. Groups U2 and U3 exhibit similarities to U1, suggesting comparable hydration products across groups at this stage. Therefore, the observed differences in appearance, morphology, microscopic morphology, and mechanical properties among the groups can be primarily attributed to the influence of steel fiber doping and sulfate erosion [44]. Comparative analysis demonstrated that a 2% volume fraction of steel fibers contributed to enhanced structural stability. However, it was observed that the detrimental effects of sulfate attack outweighed the stabilizing influence of the fibers on the structure. Figure 24 visually illustrates the erosion of UHPC by sulfate ions in sulfate solution.

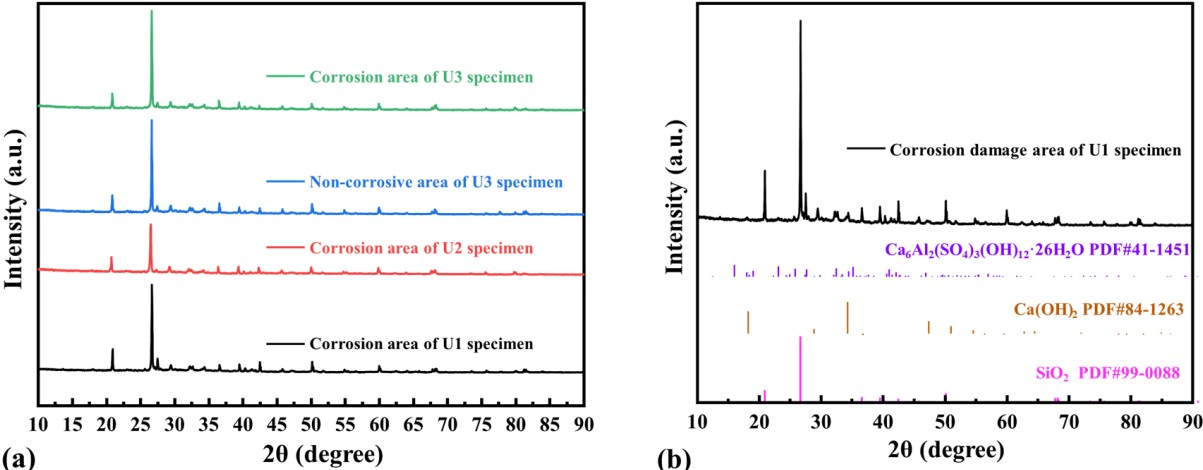

**Figure 23.** X-ray diffraction: (**a**) different conditions; (**b**) U1.

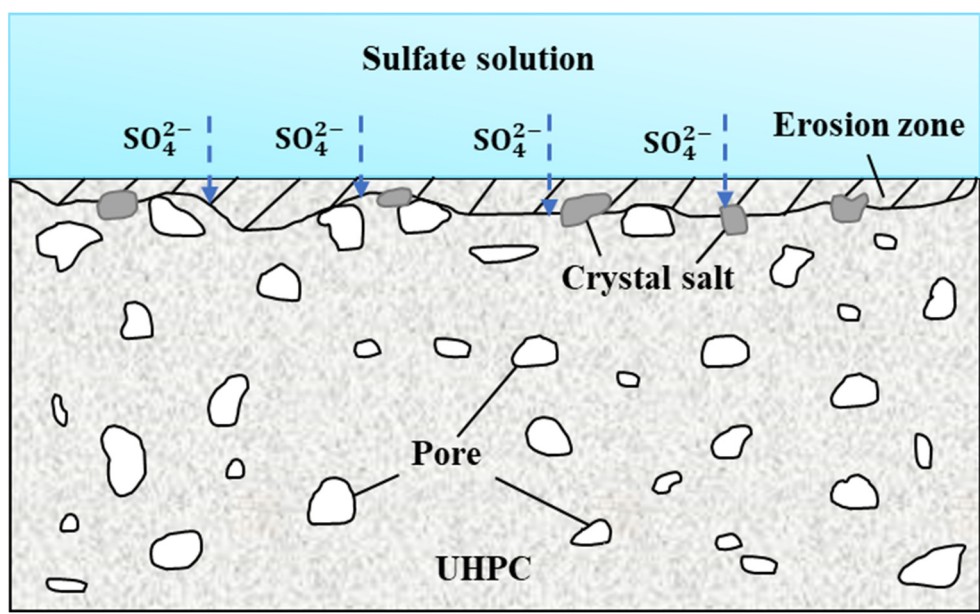

**Figure 24.** Schematic diagram of sulfate erosion.

### 5. Conclusions

In this study, the UHPC proportion design was carried out based on the theory of maximum density of particle stacking and combined with the MAA model, and its resistance to sulfate erosion was studied. The conclusions that can be drawn from this study are as follows:

(1) The study shows that when the ratio of cement to silica fume to quartz powder to quartz sand in UHPC is 716:187:416:743, with a water-to-binder ratio (W/B) of 0.20, the mechanical properties and workability of ultra-high-performance concrete (UHPC) are optimal at 3 days. At this point, the flexural strength is 9.6 MPa, the compressive strength is 55.6 MPa, and the slump flow is 227 mm. Additionally, as the volume fraction of steel fibers increases, the slump flow of UHPC decreases while its compressive and flexural strengths increase. With the addition of a 2% volume fraction of steel fibers, the mechanical properties of UHPC significantly improve and its workability remains good, with a 28-day flexural strength of 24.4 MPa, compressive strength of 132.5 MPa, and slump flow of 580 mm.

(2) Under standard curing conditions, the mass of the UHPC specimens gradually increased, with a 0.08% increase by 360 days. Under sulfate attack conditions, the mass and compressive strength corrosion resistance coefficients of two UHPC specimens, U2 and U3, showed a trend of first increasing and then decreasing, with relative mass losses of 0.73% and 0.49% at 360 days, respectively, and the compressive strength corrosion resistance coefficients decreased to 0.911 and 0.935. This is because long-term sulfate attack can induce the formation of ettringite and gypsum, promoting the growth of cracks, leading to specimen spalling, and thereby increasing the relative mass loss. It has been proven that adding fibers to the UHPC matrix can effectively reduce the mass loss caused by sulfate attack.

(3) After 360 days under sulfate attack conditions, the surface of the samples showed an increase in pore area, with these pores interconnecting to form surface cracks, and pore diameters ranging from 0.1 mm to 0.5 mm. Crystalline salts precipitated around these pores have filled some of them to a certain extent, with the pores affected by crystalline salts accounting for about 30% of the total pore proportion.

(4) The erosion layer's thickness was found to escalate over time, under conditions of standard curing and exposure to sulfate attack. At 360 days, the extent of internal damage due to sulfate erosion was approximately double that observed at 60 days.

However, when steel fibers were introduced, the escalation in damage was mitigated, with the rate of increase reducing to approximately 50% of the initial rate.

(5) After 360 days, a comparative analysis revealed morphological variances between the erosion and non-erosion zones in each group. Coupled with XRD findings, it was ascertained that the erosion zones underwent a more extensive hydration process than their non-erosion counterparts, even though both shared a similar internal material composition, with a pronounced difference in their content. UHPC specimens, subjected to standard curing, presented a high degree of hydration and structural completeness. Sulfate erosion, however, consumes these hydration products, leading to structural compromise. Importantly, the incorporation of steel fibers into the specimens markedly improves their sulfate erosion resistance.

**Author Contributions:** Conceptualization, G.W., W.C. and X.S.; data curation, G.W., W.C., X.S., J.N. and S.P.; formal analysis, G.W., W.C., X.S., J.N. and S.P.; investigation, X.R. and Y.H.; methodology, W.C., G.W. and X.S.; validation, G.W. and X.S.; resources, X.S.; visualization, G.W. and J.W.; Supervision, J.W.; funding acquisition, G.W. and J.W.; writing—original draft preparation, G.W., W.C., X.R. and X.S.; writing—review and editing, G.W., W.C. and X.R. All authors have read and agreed to the published version of the manuscript.

**Funding:** This research received no external funding.

**Institutional Review Board Statement:** Not applicable.

**Informed Consent Statement:** Not applicable.

**Data Availability Statement:** The data used to support the findings of this study are included within the article.

**Acknowledgments:** Song Xu and Yuzheng Tu helped during the experimentation and methodology; their generous support is appreciatively acknowledged.

**Conflicts of Interest:** The authors declare no conflicts of interest.

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
