# Peer review of "Enhancing Sulfate Erosion Resistance in Ultra-High-Performance Concrete through Mix Design Optimization Using the Modified Andreasen and Andersen Method"

_coatings, doi:10.3390/coatings14030274_

Round 1

Reviewer 1 Report

Comments and Suggestions for Authors

The authors present a paper focusing on the performance of UHPFRC and the sulphate erosion resistance. For this purpose, the authors indicate both in the title and in the abstract that they will use the Modified Andreasen & Andersen (MAA) method. However, the paper does not reflect this, so my first thought is that the authors should rework both the title and the abstract. In addition, throughout the paper there are several aspects that need to be improved in order to fit the requirements of this journal.

General comments.

Introduction

The state of the art is adequate, but too ambiguous with respect to the main topic of the manuscript. For example, in lines 94-95, the authors state "Despite UHPC exhibiting superior mechanical properties and durability compared to conventional concrete, research has predominantly concentrated on its mechanical characteristics, with less emphasis placed on durability aspects". I do not agree with this. It is true that the mechanical properties of UHPCs and UHPFRCs have been studied extensively. However, there are numerous papers focusing on the performance of these materials in chloride environments or against carbonation that should be mentioned. In addition, the authors should include more data in this section, indicating from the literature what improvements UHPCs represent over ordinary concretes (e.g. 90% lower porosity, etc.). This section should be improved and expanded, focusing on the main topic of the work. In addition, more focus can also be put on introducing the Modified Andreasen & Andersen (MAA) method if it is to be mentioned in the following sections.

Materials and methods

This section should definitely be structured in a more appropriate way. I think that for ease of reading, section 3.1 should be the first point in materials and methods. Explaining the Modified Andreasen & Andersen (MAA) method in depth could be useful here for the authors to introduce the objective of the work, which is not entirely clear, (i.e. to get the mixture as dense as possible). In this way, later, when they present materials, they can include a detailed table with the nomenclature of the different dosages they prepare, as it is currently not possible to know in detail the dosages. On the other hand, in the section on test methods, it should be clarified which dosages are going to undergo which tests, and the curing or exposure methods carried out, because if it is not well detailed here, when reading the results, it is very difficult to understand what has been done. For example, lines 132-133 "The raw materials required for the experiment were measured according to the predetermined design ratio", but the ratios are not specified previously.

Other remarks:

Lines 112-113, the supplier should be indicated.

Lines 145-146, the authors mention "Air voids present in the fresh UHPC were removed by 15 s of vibration". Considering the data in the literature, UHPC and UHPFRC should be considered as self-compacting concretes. Why did the authors have to include vibration after the mixing process? Why did the authors perform the slump test for self-compacting concretes, but needed to vibrate the concrete? There are several inconsistencies in the text.

Design of the UHPC Based on MAA

Deifinitely, this section should be introduced above.

3.2. W/B and Workability of the UHPC

The tittle and the content of the section is not coherent. Since it talks about compressive and flexural streght too. It should be a section of mixtures without fibres (UHPC) and another with fibres (UHPFRC) according to the tests that the authors have carried out. This whole section should also be reorganised according to the structure of the tests carried out by the authors and the mixtures analysed.

3.2. Steel fibre dosage of the UHPC (there are two sections 3.2.)

However, in this section the authors do present it as Steel fibre mixes and present other tests, much more coherent than the previous one.

4. Sulfate Attack Resistance of the UPHC

Since this section presents results, everything should be under the same section of results and discussion.

Figure 9. The legend should describe what each subgroup (U1, U2 and U3) refers to, otherwise the reader has to go backwards and it is rather awkward. Both in this and in the following ones.

4.2. Compressive strength corrosion resistance coefficient of UHPC

This section is not consistent. As I understand from the results of the compressive strength of the concrete in each test at different ages, the authors make an estimate of the corrosion resistance. But this is not the case, as it depends not only on the compressive strenght (which is possibly the least influential), but also on many other aspects. Either the authors make this much more nuanced, or they modify and approximate only by citing the influence of sulphate attack on compressive strenght. If the authors wish to include corrosion resistance in the discussion, other tests must be done.

All in all, the authors should continue to improve the work and the discussion. Conclusions must be refined according to the modifications.

Minnor comments:

Line 236, pow-der...writing error

Comments on the Quality of English Language

Minor editing of English language required

Author Response

Dear Reviewer:

Thank you for your comments on our manuscript entitled "Ultra-high Performance Concrete Proportion Design and Anti-sulfate Erosion Test Research Based on Modified Andreasen & Andersen (MAA)" Constructive comments were made.

These comments were invaluable and helped to significantly improve the quality of our paper and guided our current research work.

We have carefully studied these comments and made amendments and revisions. The revised parts have been marked in the revised manuscript. The major revisions of the paper and the point-by-point response to the reviewers' comments are given below:

Comment 1: The authors present a paper focusing on the performance of UHPFRC and the sulphate erosion resistance. For this purpose, the authors indicate both in the title and in the abstract that they will use the Modified Andreasen & Andersen (MAA) method. However, the paper does not reflect this, so my first thought is that the authors should rework both the title and the abstract.

Response: The title uses the Modified Andreasen & Andersen (MAA) method mainly to illustrate that the fit ratio of the particles of UHPC in this paper is mainly carried out by this method. In this paper, under the guidance of this method, the particle size of the raw materials was determined, and the optimal ratio of each particle component was obtained by the least squares method using MATLAB software, and then the optimal ratio of the UHPC matrix (without other phases that are not granular) was obtained by comparing the minimum content of each substance required by the specification and confirming it. Afterwards, the water consumption, additive dosage and steel fiber dosage required in this paper were screened out by the method of trial mixing, combining the workability and mechanical properties. The title of this paper has been changed to "Enhancing Sulfate Erosion Resistance in Ul-tra-High-Performance Concrete (UHPC) through Mix Design Optimization Using the Modified Andreasen & Andersen (MAA)" to more accurately express the topic.

Meanwhile, we have made additional revisions to the abstract, the outcomes of which are detailed in the manuscript.

Comment 2: For example, in lines 94-95, the authors state "Despite UHPC exhibiting superior mechanical properties and durability compared to conventional concrete, research has predominantly concentrated on its mechanical characteristics, with less emphasis placed on durability aspects". I do not agree with this. It is true that the mechanical properties of UHPCs and UHPFRCs have been studied extensively. However, there are numerous papers focusing on the performance of these materials in chloride environments or against carbonation that should be mentioned. In addition, the authors should include more data in this section, indicating from the literature what improvements UHPCs represent over ordinary concretes (e.g. 90% lower porosity, etc.). This section should be improved and expanded, focusing on the main topic of the work. In addition, more focus can also be put on introducing the Modified Andreasen & Andersen (MAA) method if it is to be mentioned in the following sections.

Response: Your comments on the durability study of UHPC are precisely what the authors need to be aware of. The article's conclusions on the durability study of UHPC are indeed problematic, which is due to the authors' lack of data collection and their own knowledge base. We have also added the advantages of UHPC over ordinary concrete, and the relevant changes will be reflected in the revised manuscript.

Comment 3: This section should definitely be structured in a more appropriate way. I think that for ease of reading, section 3.1 should be the first point in materials and methods. Explaining the Modified Andreasen & Andersen (MAA) method in depth could be useful here for the authors to introduce the objective of the work, which is not entirely clear, (i.e. to get the mixture as dense as possible). In this way, later, when they present materials, they can include a detailed table with the nomenclature of the different dosages they prepare, as it is currently not possible to know in detail the dosages. On the other hand, in the section on test methods, it should be clarified which dosages are going to undergo which tests, and the curing or exposure methods carried out, because if it is not well detailed here, when reading the results, it is very difficult to understand what has been done. For example, lines 132-133 "The raw materials required for the experiment were measured according to the predetermined design ratio", but the ratios are not specified previously.

Response: The main body of this paper consists of three sections: "Materials and Methods - Mix Design - Results and Discussion". Section 2 mainly introduces several raw materials including cement, silica fume, quartz powder, and quartz sand, as well as the experimental methods. Section 3 primarily uses the particle size information of the raw materials provided in Section 2, employs the MAA model for powder mix design, and then determines the water, admixture, and steel fiber quantities through trial mixes to establish the optimal mix proportion. Incorporating the content of Section 3 into Section 2 is not very appropriate, as this is derived from the mix ratio design steps of the third part. To clearly demonstrate the mix ratios of each UHPC specimen, we added Table 6 at the end of Section 3, which visually explains the mix ratios of specimens U1, U2, and U3.

Comment 4: Lines 112-113, the supplier should be indicated.

Response: We have included details on the manufacturers of Portland cement, silica fume, quartz sand, and quartz powder used in the revised manuscript.

Comment 5: Lines 145-146, the authors mention "Air voids present in the fresh UHPC were removed by 15 s of vibration". Considering the data in the literature, UHPC and UHPFRC should be considered as self-compacting concretes. Why did the authors have to include vibration after the mixing process? Why did the authors perform the slump test for self-compacting concretes, but needed to vibrate the concrete? There are several inconsistencies in the text.

Response: There is a problem with the statement in lines 145-146, which is due to the author's carelessness in reviewing the manuscript before submitting it, resulting in the existence of this error. As you mentioned, UHPC is a self-compacting concrete and should flow well after mixing and should not have voids. This paragraph has been revised and the details are given in the revised manuscript. Research on the workability of UHPC generally includes measuring slump, and this paper also measures the slump of our custom-mixed UHPC. This step, done before pouring, effectively shows the most realistic workability of UHPC after mixing. We believe this indicator has practical value.

Comment 6: The tittle and the content of the section is not coherent. Since it talks about compressive and flexural strength too. It should be a section of mixtures without fibres (UHPC) and another with fibres (UHPFRC) according to the tests that the authors have carried out. This whole section should also be recognized according to the structure of the tests carried out by the authors and the mixtures analysed.

Response: Section 3.2 mainly follows the previous text on designing the powder ratio of the MAA model, determining the water usage and admixture dosage. The optimal ratio is confirmed through trial mixes in various water-to-cement ratios. This step involves screening through flexural and compressive strength, where the slurry obtained after adding water, not containing steel fibers, should more appropriately be called the UHPC matrix. Subsequently, the steel fiber dosage is determined through trial mixes, followed by experiments on flexural and compressive strengths to choose the external mix ratio. These two processes are carried out step by step. The title here is not accurate enough and should be revised to "3.2 W/B of UHPC."

Comment 7: 3.2. Steel fibre dosage of the UHPC (there are two sections 3.2.)

Response: The second occurrence of 3.2 is due to the author's lack of rigorous review, and it should be corrected to "3.3 Steel Fiber Dosage of the UHPC."

Comment 8: Figure 9. The legend should describe what each subgroup (U1, U2 and U3) refers to, otherwise the reader has to go backwards and it is rather awkward. Both in this and in the following ones.

Response: Based on your suggestion, we have added explanations for U1, U2, and U3 in Figures 9 and Figures 10.

Comment 9: This section is not consistent. As I understand from the results of the compressive strength of the concrete in each test at different ages, the authors make an estimate of the corrosion resistance. But this is not the case, as it depends not only on the compressive strength (which is possibly the least influential), but also on many other aspects. Either the authors make this much more nuanced, or they modify and approximate only by citing the influence of sulphate attack on compressive strength. If the authors wish to include corrosion resistance in the discussion, other tests must be done.

Response: The reviewer's comment on the practice of characterizing corrosion resistance solely based on compressive strength is spot on. The strength of concrete is influenced by many factors, including the material's own characteristics, curing environment, among others. The approach of characterizing the corrosion resistance coefficient based on compressive strength in this paper is not an innovation of the article itself, but rather, it references existing literature. It is mainly based on the Chinese national standard GB/T 50082-2009, which uses the loss of compressive strength to represent the corrosion resistance coefficient.

Comment 10: Line 236, pow-der...writing error.

Response: We have revised this word.

We believe that these revisions have significantly improved the manuscript, making the findings clearer and the conclusions more robust. We are grateful for the opportunity to refine our work based on the insightful feedback from the reviewers.

Attached, please find the revised manuscript. We hope that our responses and the revisions adequately address the reviewers' comments.

Thank you very much for your consideration and for facilitating this process. Please do not hesitate to contact us if further information or clarification is needed.

Sincerely,

Wenlin Chen

School of Civil Engineering, Chongqing Jiaotong University, Chongqing 400074, China

622210951032@mails.cqjtu.edu.cn

Reviewer 2 Report

Comments and Suggestions for Authors

Very interesting article on UHPC.

It is well written and structured.

Information on materials and methods is well presented.

We find all the information necessary for understanding.

I noticed for figures 14 to 16, a line which passes through two points, is it really representative or interesting to trace it? because every straight line passes through two points.

Figure 17, I don't see the standard deviations.

Author Response

Dear Reviewer:

Thank you for your comments on our manuscript entitled "Ultra-high Performance Concrete Proportion Design and Anti-sulfate Erosion Test Research Based on Modified Andreasen & Andersen (MAA)" Constructive comments were made.

These comments were invaluable and helped to significantly improve the quality of our paper and guided our current research work.

We have carefully studied these comments and made amendments and revisions. The revised parts have been marked in the revised manuscript. The major revisions of the paper and the point-by-point response to the reviewers' comments are given below:

Comment 1: I noticed for figures 14 to 16, a line which passes through two points, is it really representative or interesting to trace it? because every straight line passes through two points.

Response: In this paper, the two intersecting straight lines in figures 14 to 16 are fitted using the data points from the test results, where each point represents the results of the sound wave passing through the erosion and non-erosion zones, respectively. By dividing the test block into 6 equal parts and measuring the sound wave velocity in each, the results are plotted as points in the figures. It is observed that the second point shows a significant deviation, indicating a discrepancy between the measurement in the corrosion zone and other test areas. Subsequently, the first and second points are used to fit a time-distance line representing the corrosion zone, while the first point and other points are used to fit a time-distance line for the non-corrosion zone. The correlation coefficients are found to be 0.99 or higher, aligning with established principles. This approach delineates the extent of corrosion versus non-corrosion areas based on sound wave propagation characteristics.

Comment 2: Figure 17, I don't see the standard deviations.

Response: We acknowledge the reviewer's perspective regarding the limited dataset in determining the thickness of the corroded layer shown in Figure 17. This was achieved by integrating Equations 1, 2, and 3 with the experimental results and fitted curves from Figures 14-16, where the sound velocity was directly measured and the intercepts for damaged and undamaged layers were obtained through line fitting. Since there is only a single set of data, the computed thickness is a solitary value without an average and standard deviation. We agree this is an area for improvement through additional experiments in follow-up studies, which would provide statistical parameters. However, owing to constraints on time, supplementing the current paper with further tests is unfeasible. We appreciate this constructive critique to strengthen our analysis approach moving forward.

We believe that these revisions have significantly improved the manuscript, making the findings clearer and the conclusions more robust. We are grateful for the opportunity to refine our work based on the insightful feedback from the reviewers.

Attached, please find the revised manuscript. We hope that our responses and the revisions adequately address the reviewers' comments.

Thank you very much for your consideration and for facilitating this process. Please do not hesitate to contact us if further information or clarification is needed.

Sincerely,

Wenlin Chen

School of Civil Engineering, Chongqing Jiaotong University, Chongqing 400074, China

622210951032@mails.cqjtu.edu.cn

Reviewer 3 Report

Comments and Suggestions for Authors

The article introduces an experimental and analytical study to evaluate the effect of sulfate attack on mechanical properties and erosion resistance of ultra-high-performance concrete mixtures. The article can be considered suitable for publication after considering the following recommendations.

1-     The abstract should be a stand-alone summary of the whole article, where it should briefly introduces the main elements of the work including its aim, methodology, and major outputs. The abstract of this article lags most of these elements, where it mentions nothing about the methodologies used including the experimental work, while jumped directly to the obtained results. The abstract must be completely reformulated to meet the above-mentioned requirements. The experimental parameters and tests must be addressed in sufficient details but briefly. The most important numeral results and comparisons must also be reported at the end of the abstract section.

2-     The last paragraph of the introduction does not clearly mention specified new contributions of this work over the existing similar research work, where only a general statement is addressed at the start of the paragraph. The novel points of the work must explicitly be clarified at the end of the introduction section.

3-     On what basis the erosion was calculated using Equations 1 to 3? More details are required.

4-     A unified table that lists all mixtures with material quantities must be introduced in section 2 instead of presenting different isolated tables in section 3. It is not clear how many mixtures were prepared and what the investigated variables are. These details must be given in section 2 not in section 3.

5-     Figure 6 has no legend so that readers cannot distinguish the bars of compressive strength from those of flexural strength. The same issue stands for Figure 7.

6-     The conclusion points related with the workability and mechanical properties and their analyses (conclusions 1 to 3) must be supported by numeral comparisons.

Author Response

Dear Reviewer:

Thank you for your comments on our manuscript entitled "Ultra-high Performance Concrete Proportion Design and Anti-sulfate Erosion Test Research Based on Modified Andreasen & Andersen (MAA)" Constructive comments were made.

These comments were invaluable and helped to significantly improve the quality of our paper and guided our current research work.

We have carefully studied these comments and made amendments and revisions. The revised parts have been marked in the revised manuscript. The major revisions of the paper and the point-by-point response to the reviewers' comments are given below:

Comment 1: The abstract should be a stand-alone summary of the whole article, where it should briefly introduces the main elements of the work including its aim, methodology, and major outputs. The abstract of this article lags most of these elements, where it mentions nothing about the methodologies used including the experimental work, while jumped directly to the obtained results. The abstract must be completely reformulated to meet the above-mentioned requirements. The experimental parameters and tests must be addressed in sufficient details but briefly. The most important numeral results and comparisons must also be reported at the end of the abstract section.

Response: Thank you for your valuable suggestions. We have made detailed revisions to the abstract section based on your advice, and the specific details can be found in the newly revised manuscript.

Comment 2: The last paragraph of the introduction does not clearly mention specified new contributions of this work over the existing similar research work, where only a general statement is addressed at the start of the paragraph. The novel points of the work must explicitly be clarified at the end of the introduction section.

Response: In response to your suggestion, we have added a section on the innovation and novelty of this study to the last paragraph of the introduction.

Comment 3: On what basis the erosion was calculated using Equations 1 to 3? More details are required.

Response: Based on the equal time used for ultrasonic diffraction propagation and refraction propagation, we can derive Equation (1). By converting Equation (1), we can obtain Equation (2) and Equation (3).

Comment 4: A unified table that lists all mixtures with material quantities must be introduced in section 2 instead of presenting different isolated tables in section 3. It is not clear how many mixtures were prepared and what the investigated variables are. These details must be given in section 2 not in section 3.

Response: The main body of this paper consists of three sections: "Materials and Methods - Mix Design - Results and Discussion". Section 2 mainly introduces several raw materials including cement, silica fume, quartz powder, and quartz sand, as well as the experimental methods. Section 3 primarily uses the particle size information of the raw materials provided in Section 2, employs the MAA model for powder mix design, and then determines the water, admixture, and steel fiber quantities through trial mixes to establish the optimal mix proportion. Based on the optimal mix ratio, three types of test blocks are prepared for the study of sulfate attack: one set without fibers and always in standard curing conditions, one set without fibers placed in a sulfate attack environment, and one set with 2% fibers added and placed in a sulfate attack environment. The mechanical properties, appearance morphology, corrosion layer thickness, and microstructure of the three groups of test blocks are tested at 60, 120, 180, 240, 300, and 360 days, and the samples' susceptibility to sulfate attack is comprehensively analyzed. Incorporating the content of Section 3 into Section 2 is not very appropriate, as there indeed exists the phenomenon of scattered tables in Section 3. The author will pay more attention to this issue in future writings.

Comment 5: Figure 6 has no legend so that readers cannot distinguish the bars of compressive strength from those of flexural strength. The same issue stands for Figure 7.

Response: We have added the corresponding legends to Figures 6 and 7; the details of the changes can be found in the manuscript.

Comment 6: The conclusion points related with the workability and mechanical properties and their analyses (conclusions 1 to 3) must be supported by numeral comparisons.

Response: In response to your feedback, we've revised the conclusions of the article to include quantitative descriptions. The details of the revisions can be found in the conclusion section of the paper.

We believe that these revisions have significantly improved the manuscript, making the findings clearer and the conclusions more robust. We are grateful for the opportunity to refine our work based on the insightful feedback from the reviewers.

Attached, please find the revised manuscript. We hope that our responses and the revisions adequately address the reviewers' comments.

Thank you very much for your consideration and for facilitating this process. Please do not hesitate to contact us if further information or clarification is needed.

Sincerely,

Wenlin Chen

School of Civil Engineering, Chongqing Jiaotong University, Chongqing 400074, China

622210951032@mails.cqjtu.edu.cn

Round 2

Reviewer 1 Report

Comments and Suggestions for Authors

The authors have improved the paper based on the reviewers' comments.